

# A generalized photon-tracking approach to simulate spectral snow albedo and transmissivity using X-ray microtomography and geometric optics

Theodore Letcher[1], Julie Parno[1], Zoe Courville[1], Lauren Farnsworth[1], and Jason Olivier[1]

[1]US Army Corps of Engineers Cold Regions Research and Engineering Laboratory,Hanover, NH

**Correspondence:** Theodore Letcher (Theodore.W.Letcher@erdc.dren.mil)

**Abstract.** A majority of snow radiative transfer models (RTM) treat snow as a collection of idealized grains rather than an organized ice-air matrix. Here we present a generalized multi-layer photon-tracking RTM that simulates light transmissivity and reflectivity through snow based on X-ray microtomography, treating snow as a coherent structure rather than a collection of grains. Notably, the model uses a blended approach to expand ray-tracing techniques applied to sub-1 cm$^3$ snow samples to snowpacks of arbitrary depths. While this framework has many potential applications, this study's effort is focused on simulating light transmissivity through thin snowpacks as this is relevant for surface energy balance applications and sub-nivean hazard detection. We demonstrate that this framework capably reproduces many known optical properties of a snow surface, including the dependence of spectral reflectance on snow grain size and incident zenith angle and the surface bidirectional reflectance distribution function (BRDF). To evaluate how the model simulates transmissivity, we compare it against spectro-radiometer measurements collected at a field site in east-central Vermont. In this experiment, painted panels were inserted at various depths beneath the snow to emulate thin snow. The model compares remarkably well against the spectroradiometer measurements. Sensitivity simulations using this model indicate that snow transmissivity is greatest in the visible wavelengths and is limited to the top 5 cm of the snowpack for fine-grained snow, but can penetrate as deep as 8 cm for coarser grain snow. An evaluation of snow optical properties generated from a variety of snow samples suggests that coarse grained low density snow is most transmissive.

## 1 Introduction

Due to the highly reflective nature of snow, seasonal snowpacks make the surface significantly more reflective when present, impacting regional weather and climate. Correspondingly, the snow albedo feedback, caused by changes in seasonal snow cover extent and properties, represents one of the more dramatic markers of regional and global climate change (e.g., Hall, 2004; Déry and Brown, 2007; Flanner et al., 2011; Letcher and Minder, 2015; Thackeray and Fletcher, 2016). While snow is highly reflective, snow albedo is not equal for all snowpacks. For instance, snow albedo typically decreases with snow age due



to metamorphic processes resulting in larger snow grains (e.g., Wiscombe and Warren, 1980; Aoki et al., 2000; Flanner and Zender, 2006; Adolph et al., 2017). Snow albedo is also diminished by light absorbing impurities such as dust or black carbon

that contaminate the snow (e.g., Doherty et al., 2010; Painter et al., 2012; Skiles et al., 2012; Dumont et al., 2014; Skiles et al., 2015; Skiles and Painter, 2019; Shi et al., 2021). Importantly, these two effects impact different parts of the electromagnetic spectrum, with grain size having a greater influence in the near-infrared (NIR), and the particle contamination influencing the visible region. Finally, even though thin snow covers are highly reflective, the aggregate surface albedo for a thin snow cover can be influenced by the underlying ground surface depending on the snow microstructure (Perovich, 2007; Warren,

2013; Libois et al., 2013). Understanding these small-scale drivers of snow albedo is important for large-scale remote sensing applications and regional weather and climate modeling.

There are several documented approaches to model snow broadband and spectral albedo using radiative transfer models (RTMs) in efforts to better understand and predict the effects of snow aging and impurities on snow optical properties. While a full review of snow radiative transfer is well beyond the scope of this paper, we refer the reader to He and Flanner (2020)

for a rigorous overview of the different approaches. There are also numerous simplified parameterizations for snow albedo of varying complexity designed for implementation in weather and climate models (e.g., Verseghy, 1991; Dickinson, 1993; Gardner and Sharp, 2010; Vionnet et al., 2012; Saito et al., 2019; Bair et al., 2019).

At a fundamental level, the scattering of electromagnetic energy incident upon the boundary separating a snow grain and the surrounding air is determined by the different refractive indices for ice and air. The absorption of light as it passes through solid

ice is well understood and has a strong wavelength dependence (e.g., Grenfell and Perovich, 1981; Perovich and Govoni, 1991; Warren and Brandt, 2008). The scattering of visible and NIR light at an air/ice boundary is well described by the geometric optics approximation, which requires that the wavelength of light is small relative to the size of the scattering particle. While the physics behind scattering and absorption are well understood for a single snow particle, the actual path of a light ray through a snowpack can be extraordinarily convoluted as the ray is constantly intersecting air/ice interfaces with very little absorption.

Seminal studies describing snow albedo modeling (e.g., Warren and Wiscombe, 1980; Wiscombe and Warren, 1980) and most subsequent approaches treat snow grains as independent scatterers, where the scattering properties of an individual grain are not affected by adjacent grains and are independent of the spacing between grains and, thus, snow density. Mie theory is typically used to calculate snow albedo owing to its computational efficiency (e.g., Bohren and Beschta, 1979; Wiscombe, 1980). However, Mie theory is most often applied to particles with spherical geometries, and so it is common practice to

represent snow by a collection of effective spheres. Yet although snow grain size is often most cited as the key driver of pure snow albedo, grain shape also has an impact on snow spectral albedo (Aoki et al., 2000; Libois et al., 2013; Dang et al., 2016). For instance, an inter-comparison of several commonly-used two stream RTMs parameterizing snow grains as equivalent spheres conclude that these models overestimate snow albedo in the NIR (Dang et al., 2019).

Efforts to understand and simulate the impacts of snow particle shape on snow spectral albedo have largely focused on

leveraging the geometric optics approximation in various ways. For instance, Yang and Liou (1996) used ray-tracing to compute the single scattering properties of idealized hexagonal columns, plates, and rosettes. Grundy et al. (2000) presented a Monte Carlo approach to estimate optical properties of computer-rendered 3D spheres that compared well with Mie theory. Their





work was extended to estimate the scattering properties of irregularly shaped crystals. A more recent effort by Xiong et al.
(2015) focused on determining the optical properties of an idealized mixed snow/air medium generated from a randomized

bicontinuous 2D representation of the snow. More recently, several studies have used ray-tracing and photon-tracking methods
to simulate snow optical properties from renderings of snow particles generated from X-ray microtomography (hereby: $\mu$CT)
scans of real snow (Haussener et al., 2012; Kaempfer et al., 2007; Ishimoto et al., 2018; Dumont et al., 2021).

Collectively, RTM-focused studies of snow have greatly expanded the knowledge surrounding the optical properties of
irregular snow grains and informed the role of snow microstructure on spectral reflectance. However, the impact of snow

microstructure on snow spectral transmissivity has often been overlooked in observations and modeling.

In this study, we build upon the approaches of Grundy et al. (2000), Kaempfer et al. (2007), Jacques (2010), and Xiong
et al. (2015) to develop a Monte Carlo photon-tracking snow RTM that is driven by $\mu$CT observations of snow and is designed
for broad applications, including snow transmissivity. The primary purpose of this RTM framework is to use explicit photon-
tracking techniques in conjunction with 3D renderings of snow samples to estimate realistic snow optical properties for use

in a Monte Carlo photon-tracking model. In section 2 we describe the model framework and $\mu$CT data processing. In section
3, we demonstrate the model's capability to reproduce known optical properties of snow, compare model output to spectral
albedo measurements of objects buried beneath snow at various depths, and use the RTM to investigate snow transmissivity. In
sections 4 and 5, we present a broad discussion and conclusions.

## 2  Data and Methods

Here we describe the framework of a new, semi-quantized Monte Carlo photon-tracking model designed to simulate radiative
transfer (RT) through snow with a focus on spectral albedo and transmissivity in the visible and NIR (i.e., $380 \leq \lambda \leq 1300$
nm). While computationally expensive, there are several advantages to the Monte Carlo approach over more traditional ap-
proaches that aim to solve the radiative transfer equation (RTE). In particular Monte Carlo models are useful for modeling RT
through non-spherical particles and for 3D RT applications (e.g., Iwabuchi, 2006; Whitney, 2011). In this model, the Monte

Carlo approach is used in order to eliminate all assumptions regarding snow microstructure. Specifically, this approach treats
the snowpack more as a coherent structural ice lattice rather than as a collection of idealized particles. The explicit photon-
tracking through 3D renderings of snow performed as part of this model are similar to those described in Grundy et al. (2000);
Kaempfer et al. (2007); Dumont et al. (2021); Ishimoto et al. (2018). Additionally, the Monte Carlo approach lends itself well
to parallelization, and the semi-quantized approach described here reduces the number of photons required to achieve a statis-

tically robust result. Yet, this approach is not without its drawbacks. For instance, the geometric optics approximation localizes
the ray, directing all scattered radiation along a single path. Further, this framework ignores the wave properties of light, such
as phase and diffraction, which limits its overall applicability and reduces accuracy. Despite these drawbacks, numerous ap-
proaches in the literature have demonstrated success in simulating snow reflectance of natural (i.e., unpolarized) light using
these simplifications (e.g., Kaempfer et al., 2007; Malinka, 2014; Xiong et al., 2015).



This model can be divided into two distinct components. The first of which determines key snow medium optical properties by launching photons into 3D closed-surface renderings of snow samples derived from $\mu$CT scans with a voxel resolution of $\approx 20$ $\mu$m. The second uses the optical properties derived from the first part to drive a 1D photon-tracking model whereby individual photon packets are prescribed a random initial position and incident direction on the snow. Each individual photon packet then has a unique path whereby all of the energy contained within a given packet travels in the same direction and the

amount of energy within a given packet is depleted continuously according to absorption within the medium. Note that for both model components, the ice refractive indices reported by Warren and Brandt (2008) are used to compute scattering and absorption.

## 2.1 Snow Optical Properties

The 1D medium model requires three key optical properties: the extinction coefficient ($\gamma_{ext}$), the mean path fraction traveled

within ice ($F_{ice}$), and the scattering phase function ($p(cos\Theta)$). In considering light as a ray traveling through the snow medium, which is scattered each time it intersects an air/ice boundary and partially absorbed within the ice, the extinction coefficient is related to the distance traveled between scattering and absorption events. The phase function determines the change in direction of the ray during a scattering event, and the ice-path fraction, when combined with the ice absorption coefficient, determines the mean energy depleted from the ray for a given distance traveled between scattering events. For a given snow sample,

$\gamma_{ext}$ is determined following the method described in Xiong et al. (2015) applied to the 3D rendering of snow as opposed to an idealized bicontinuous medium. In this framework, photons are initialized at a random position within the snow sample, and launched in a random direction for a specified distance ($L$). If the photon is initialized within the air, the probability of extinction ($P_{ext}$) is 1 if a boundary is intersected over $L$, otherwise it is 0. In the case where the photon is initialized within the ice medium, $P_{ext}$=1 if a boundary is intersected over $L$, otherwise it is given as:

$$P_{ext} = 1 - e^{-\kappa_\lambda L}, \tag{1}$$

where $\kappa_\lambda$ is the wavelength-dependent absorption coefficient of ice, which is related to the imaginary part of the ice refractive index ($k$):

$$\kappa_\lambda = \frac{4\pi k}{\lambda}. \tag{2}$$

This slight modification is made to account for the added probability of extinction due to absorption of the photon within the

ice particle. Note that this also introduces a minor wavelength dependence into the extinction coefficient. Using this method, a probability of extinction can be determined for distance $L$. This method is repeated for several distances ranging from the voxel resolution (20 $\mu$m) to the width of the snow sample volume (e.g., 10 mm). The extinction coefficient is then determined using a curve fit to Beer-Lambert law:





$$P_{ext} = 1 - e^{-\gamma_{ext}L},\qquad(3)$$

The mean fractional ice path ($F_{ice}$) is determined by tracking individual photons as they travel throughout the aggregate snow sample. This framework closely mimics that of Kaempfer et al. (2007) in that photons travel through the snow medium and change direction according to Snell's law of refraction and a probabilistic representation of Fresnel's law of reflectance. Here, a photon is initialized at a random starting point somewhere within the snow sample and launched in a random direction. The photon is tracked until it exits the medium, and the $F_{ice}$ is simply the ratio of the distance traveled within ice over the total distance traveled. This is repeated for a large number of photons to determine an average $F_{ice}$.

Fresnel's law dictates that the fractional reflection and transmission of light at a boundary is related to the incident angle ($\theta_i$) and the refractive indices ($n$) of the two media separated by the boundary:

$$R_h = \frac{n_1 cos\theta_i - n_2 \sqrt{1 - \left(\frac{n_1}{n_2} sin\theta_i\right)^2}}{n_1 cos\theta_i - + n_2 \sqrt{1 - \left(\frac{n_1}{n_2} sin\theta_i\right)^2}},\qquad(4)$$

and

$$R_v = \frac{n_1 \sqrt{1 - \left(\frac{n_1}{n_2} sin\theta_i\right)^2} - n_2 cos\theta_i}{n_1 \sqrt{1 - \left(\frac{n_1}{n_2} sin\theta_i\right)^2} + n_2 cos\theta_i},\qquad(5)$$

where $R_h$ and $R_v$ are the horizontally and vertically polarized reflectances. Assuming that the radiation is unpolarized (e.g., natural light), the reflectance ($R$) is:

$$R = \frac{1}{2}\left(R_h^2 + R_v^2\right).\qquad(6)$$

Through energy conservation, the transmittance ($T$) is simply:

$$T = 1 - R.\qquad(7)$$

Then, if the vector normal to the boundary plane ($\hat{v_n}$) is oriented towards the medium with refractive index $n_1$, the direction unit vectors for transmitted and reflected radiation are computed as:

$$\hat{v_r} = \hat{v_i} + 2cos\theta_i \hat{v_n}\qquad(8)$$





and

$$\hat{v}_t = \frac{n_1}{n_2}\hat{v}_i + \left(\frac{n_1}{n_2}cos\theta_i - cos\theta_t\right)\hat{v}_n, \tag{9}$$

where $\hat{v}_r$ and $\hat{v}_t$ are the reflection and transmission unit direction vectors, respectively.

The phase function is determined by separating out individual snow grains from the reconstructed 3D snow sample and probing them with photons to compute the scattering angle where the cosine of the scattering angle ($\Theta$) is the dot product between the directional unit vector of radiation incident on the particle ($\hat{\Omega}'$) and the directional unit vector of the scattered radiation ($\hat{\Omega}$) in the cartesian coordinate space:

$$cos\Theta = \hat{\Omega}' \cdot \hat{\Omega}. \tag{10}$$

The phase function is constructed by first initializing a photon outside of a given particle and firing it in a random direction towards the particle. The photon then interacts with the snow particle, guided by Eqs. 4 - 9. For each collision ($i$), the amount of energy exiting the particle into the directional bin ($cos\Theta_j$) is tracked, and the remaining photon energy is depleted as energy exits, or is absorbed within, the ice particle. This is repeated for an arbitrary number of collisions ($n$) until there is less than 0.1% of the initial energy left (Fig. 1). For the initial collision ($i = 0$), the energy exiting the particle is simply the reflected fraction of the incident ray, and for each subsequent collision, it is the transmitted multiplied by the remaining ray energy (e.g., Malinka, 2014):

$$W_{\Theta_i} = \begin{cases} R_0 & i = 0 \\ T_0 T_i & i = 1 \\ T_0 \left(\prod_{n=1}^{i-1} R_n e^{-\kappa_\lambda s_n}\right) T_i & i > 1 \end{cases}, \tag{11}$$

where $s_n$ is the distance traveled within the particle between the boundary intersections $n-1$ and $n$. For visible wavelengths, the impact of absorption on the phase function is negligible, however for wavelengths exceeding 1000 nm, it becomes more important. The resulting distribution of energy is converted to a phase function defined relative to the ray initially incident upon the scattering particle following Grundy et al. (2000):

$$p(cos\Theta_j) = \frac{4\pi N_j}{N sin\Theta_j d\Theta}, \tag{12}$$

where $N$ is the total photon energy and $N_j$ is the total photon energy directed into bin $j$.

In practice, while a majority of photons require only a few collisions to reach the 0.1 % energy threshold, we cap the number of internal collisions to a maximum of 10 to limit both computation time and the accumulation of error caused by imperfect



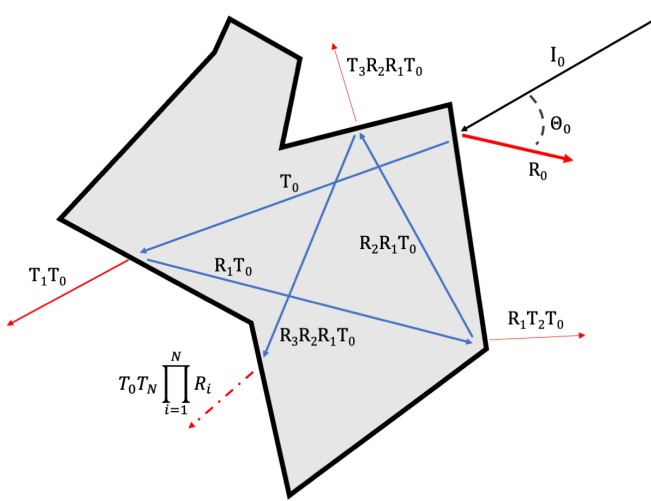

**Figure 1.** Schematic illustration of the photon-tracking method used to determine the scattering phase function for a grain with an arbitrary shape.

3D particle representations. To evaluate this method, we use it to estimate the phase function of a collection of idealized crystal habits, including spheres, hexagonal plates, and columns with a size parameter of 1000 (Fig. 2). We find that these phase

functions are in good agreement with Mie theory for spheres, and with phase functions reported for other shapes presented in previous studies focused on the scattering properties of spherical, idealized, and irregular snow crystal shapes (Iaquinta et al., 1995; Macke et al., 1996; Yang and Liou, 1996; Grundy et al., 2000; Malinka, 2014; Ishimoto et al., 2018). To determine the phase function for 1D model, this ray-tracing method is applied to a selection of rendered grains from the $\mu$CT sample, and $p(cos\Theta_j)$ is computed for each bin following eq. 12 where $N_j$ is determined from all selected grains. Note that in this method

the selected grains are checked to ensure that they contain no facets touching the sample boundaries.

## 2.2  1D photon-tracking Model

Once the required optical properties of the snow sample are determined by launching photons through $\mu$CT sample volumes, a 1D photon-tracking model is used to simulate snow spectral albedo, transmissivity, and Bidirectional Reflectance Distribution Function (BRDF). The 1D model is used in place of the explicit photon-tracking model described by Kaempfer et al.

(2007) in order to allow for the computationally feasible simulation of spectral albedo and transmissivity for snow covers with depths exceeding 1 cm with sufficient grain resolution. Additionally, it is used complications associated with lateral boundary treatment and stitching multiple $\mu$CT scans together into a coherent snow lattice. Our 1D model is based largely on Jacques (2010), which describes a semi-random 1D multi-layer Monte Carlo photon-tracking approach for application in the field of biomedical imaging. In this framework, discrete, plane-parallel, snow layers with optical properties constant throughout each

layer are first prescribed. Then a photon packet is initialized at some starting position ($\boldsymbol{X_0}$) with cartesian components of ($x_0$,


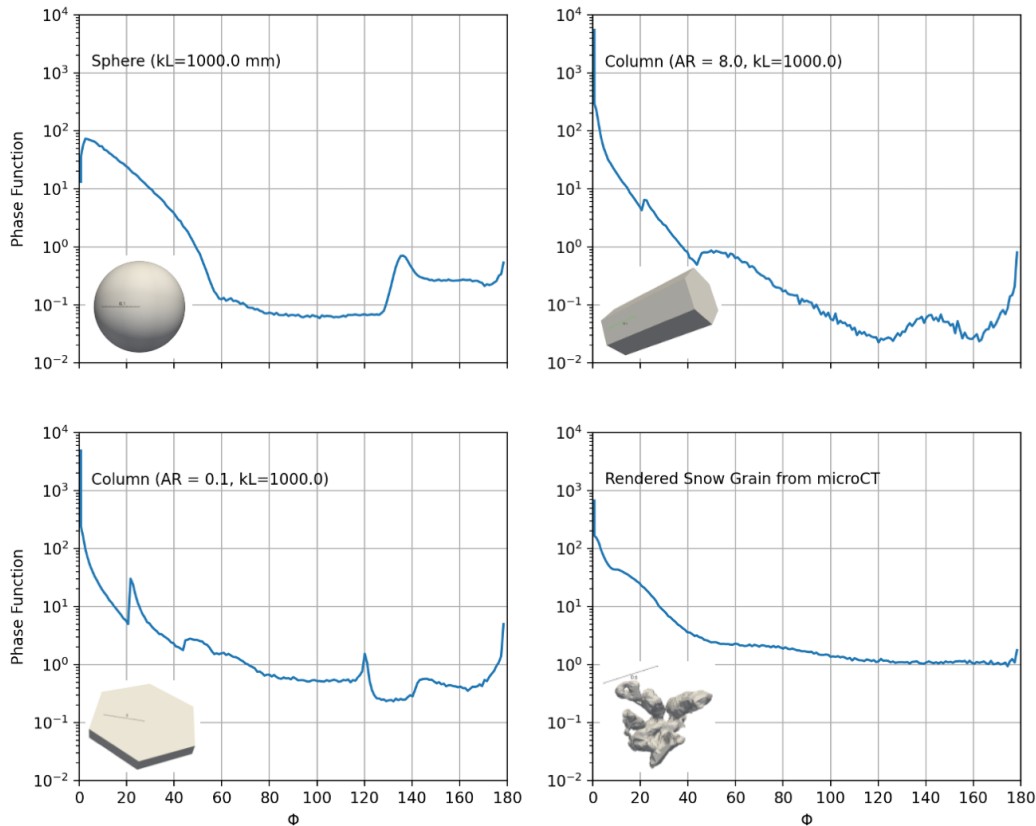

**Figure 2.** Phase functions for idealized snow particles following equations 11 and 12 for incident radiation at $\lambda$=900nm. The size parameter ($kL$) for each particle is set to 1000. Note that for the rendered grain, the approximate grain diameter yielded a size parameter of approximately 1400. 500,000 photons were used to generate the phase function. AR is the axis ratio between the long "c" and short "a" axes of the hexagonal column.

$y_0$, $z_0$) and an initial energy of unity ($E = 1$). An initial unit direction vector for the photon is given in cartesian coordinates as:

$$V_0 = \left[ sin\theta cos(\phi), sin(\theta) sin(\phi), -cos\theta \right],\tag{13}$$

where $\theta$ is the solar zenith angle, and $\phi$ is the azimuth angle clockwise from $x$. This initial direction can be prescribed randomly

(i.e., diffuse radiation), or at any specified zenith/azimuth angle (i.e., direct radiation), or as a mixture of both diffuse and direct radiation.

Once the initial position is set, the photon is launched into the medium, and travels a distance $s$ before experiencing a scattering event. $s$ is computed statistically using the Beer-Lambert law and the medium extinction coefficient (Jacques, 2010):





$$s = -\frac{ln\zeta}{\gamma_{ext}}, \tag{14}$$

where $\zeta$ is a random uniform number between 0 and 1. The new position in the medium is:

$$\boldsymbol{X} = \boldsymbol{X_0} + s\boldsymbol{V_0} \tag{15}$$

At the scattering event, the photon packet is given a new direction unit vector according to the scattering phase function. Because this framework treats the scattering phase function as a probability distribution function (PDF), the scattering angle $\Theta$ is determined by choosing a random sample from $p(cos\Theta)$ PDF:

$$P(cos\Theta) = \frac{p(cos\Theta)d\Omega}{4\pi}, \tag{16}$$

where $P$ is the probability of light being scattered into a cone with solid angle $d\Omega$ in the direction $\Theta$ from the incident radiation given the phase function.

Then the new direction vector is determined from $\Theta$ (Jacques, 2010):

$$\begin{aligned} \mu_x &= \frac{sin\Theta\left(\mu_{x_0}\mu_{z_0}cos\phi - \mu_{y_0}sin\phi\right)}{\sqrt{1-\mu_{z_0}^2}} \\ \mu_y &= \frac{sin\Theta\left(\mu_{y_0}\mu_{z_0}cos\phi - \mu_{x_0}sin\phi\right)}{\sqrt{1-\mu_{z_0}^2}} \\ \mu_z &= -\sqrt{1-\mu_{z_0}^2}\,sin\Theta\cos\phi + \mu_{z_0}cos\Theta \end{aligned}, \tag{17}$$

where $\phi$ is given as a uniform random number between 0 and $2\pi$, the 0 subscript represents the incident direction, and $\mu_x$, $\mu_y$, and $\mu_z$ make up the components of the unit direction vector.

Photon energy is depleted over distance $s$ according to the ice absorption coefficient and $F_{ice}$ as determined from the $\mu$CT data instead of using a medium absorption coefficient:

$$E = E_0\left(e^{-\kappa_\lambda s F_{ice}}\right), \tag{18}$$

where $E$ is the new photon energy, and $E_0$ is the incident photon energy.

To achieve statistical energy conservation, a "Russian Roulette" function is used to determine whether or not to fully absorb (i.e., kill) the photon packet once its energy falls below a prescribed threshold (Iwabuchi, 2006; Jacques, 2010). This is given as:

$$E = \begin{cases} mE & \zeta \leq 1/m \\ 0 & \zeta > 1/m \end{cases}, \tag{19}$$





where $\zeta$ is a random number between 0 and 1, and $m$ is a prescribed constant on the order of 1-10. By treating absorption continuously rather than probabilistically, the number of photons required to attain a robust solution is significantly reduced, and further ensures that the model cannot get stuck in an infinite loop.

If the $z$ position of a photon-packet is above the top of the snow surface (i.e., it has exited the top of the snowpack), the remaining energy within the packet is added to the total reflected energy and the photon is eliminated. In an open lower-
boundary configuration, if a photon-packet $z$ position is less than 0 (i.e., it has exited the bottom of the snowpack) the remaining energy is added to the total transmitted energy, and the photon is eliminated. Alternatively, a lower boundary can be simulated with a specified spectral reflectance such that a portion of the photon energy will be absorbed at the lower boundary, and the remaining energy will be reflected upward. Once all photons have been eliminated from the model, the simulation is complete.

This model is extended to a multilayer configuration, by simply defining unique optical properties corresponding to specified
depths throughout the snowpack. When a photon packet travels from one layer to another, its trajectory and energy depletion are determined by the optical properties of the new layer.

The basic premise of this model is illustrated in figure 3, which traces the position and energy of two photons on a 2D plane as they travel throughout an idealized two-layer snowpack 10 cm deep.

### 2.3 Directional Conic Reflectance Function

The reflectance of a surface is often described using the concept of a BRDF (e.g., Stamnes and Stamnes, 2016). This concept essentially represents a PDF of reflected direction of a ray of light impacting the surface from a given incident direction, and is used to simplify the complex reflectance properties of a rough surface (e.g., shading and multiple reflections). To estimate the BRDF from this model, we follow the methods described in Kaempfer et al. (2007). In this framework, the BRDF for specified incident zenith and azimuth directions is approximated using the Directional Conic Reflectance Function (DCRF),
which computes the energy reflected into a cone in the direction: $\theta_r, \phi_r$ subtended by solid angle: $d\Omega$:

$$DCRF(\theta_i, \phi_i, \theta_r, \phi_r) = \frac{I_r(\theta_r, \phi_r)}{I_i(\theta_i, \phi_i) cos\theta_i d\Omega}, \tag{20}$$

where $I$ is the radiative flux, and the subscripts $i$, and $r$ correspond to the incident and reflected radiation, respectively.

### 2.4 Snow sampling and spectroradiometer measurements in the field

To evaluate the model, we collected snow samples and snow surface spectroradiometer measurements at Union Village Dam
(UVD) in Thetford, Vermont several times throughout the 2020-21 winter. The UVD site is a broad flat clearing surrounded by deciduous forests spanning approximately 40000 $m^2$, and bounded on the southern end by the Ompompanoosuc River. During each data collection, a snow pit was excavated and standard snow characteristics, such as snow depth, density, and grain size were measured manually. Several snow samples were carefully extracted in columns adjacent to the snow pit sidewalls in cylindrical containers 7 cm high x 1.9 cm in diameter (Fig. 4). These samples were transported in a hard, plastic cooler for 10

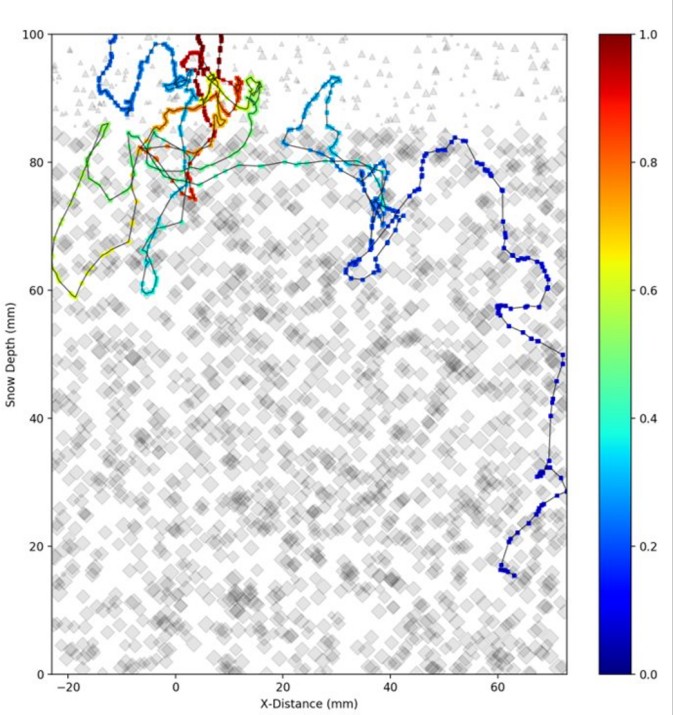

**Figure 3.** x/z cross section of two photons within a multi-layered snowpack 10 cm deep. The color scale indicates the fractional energy of the photon packet. The gray markers aid in showing the different snow layers. Note that only one photon exits the top of the snowpack, the other is fully absorbed in the lower layer. Note that the markers are provided as a visual aid, and not representative of the snow particle shape or distribution.

miles from the UVD site to the Cold Regions Research and Engineering Laboratory (CRREL). At CRREL they were stored in a -20 °C cold room prior to $\mu$CT analysis. These samples were not casted (i.e. not preserved using a pore-filler).

 Spectral reflectance and transmissivity data were collected using a Malvern Panalytical ASD FieldSpec 4 Hi-Res: High Resolution Spectroradiometer. The FieldSpec 4 has a spectral range of 350-2500 nm and a spectral resolution of 3 nm in the visible and 10 nm in the SWIR. The data collection was performed within 1.5 hours of solar noon in order to limit high zenith
angle impacts. An optimization was conducted prior to the start of data collection and any time lighting conditions changed in order to ensure accurate reflectance readings. Data collections were taken 2.5 to 3 feet above the snow surface using a 5 degree field of view optic lens, resulting in a measurement footprint diameter of approximately 6 cm. The collection strategy employed included taking a white reference reading from a pure reflective panel and five readings at different locations on the target surface; the mean of the five readings was used as the reflectance value for that specific location.

In this paper, we focus specifically on data collected on 12 February, 2021 as this day had the most stable ambient lighting conditions and resulted in the majority of our snow and reflectance measurements. At the time of the measurements the sky was covered with a high optically thick overcast, and as a result the ambient lighting conditions were generally diffuse. The




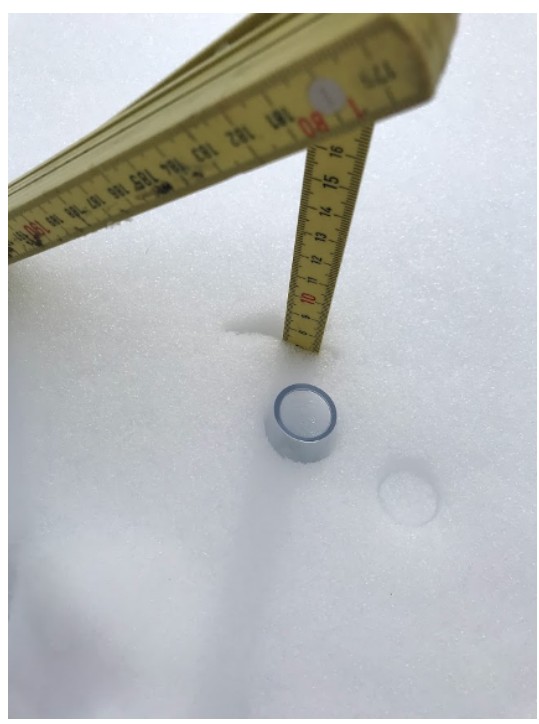

**Figure 4.** Photograph showing snow sample collection for $\mu$CT analysis.

snow was dry and approximately 34 cm deep, and was roughly characterized as a layer of relatively fresh snow approximately 10 cm deep overlying a layer comprised of larger mixed refrozen snow grain clusters and facets, separated by a 1 cm thick ice
crust. To simulate the effects of a shallow snowpack, an aluminum panel painted black was inserted into the snowpack through the snow pit sidewall at three depths (10 cm, 4.75 cm, and 2.5 cm) with care as not to damage the smooth snow surface (e.g., Fig. 5). This panel was strongly absorptive in the visible and NIR spectrum with a constant reflectance of approximately 4 % throughout the entire 350 - 2500 nm spectrum. Since there was no appreciable difference between the measured spectral albedo of the virgin snow (i.e., no inserted panel) and the panel inserted at 10 cm, we limit our analysis to the 4.75 and 2.5 cm panel
depths.

## 2.5   $\mu$CT sampling and Mesh Generation

These snow samples were characterized at the microscale with a cold-hardened Bruker Skyscan 1173 $\mu$CT scanner housed in a -10 °C cold room equipped with a Hamamatsu 130/300 tungsten X-ray source, which produces a fixed conical, polychromatic beam with a spot size of <5 $\mu$m and a flat panel sensor camera detector. Each sample was scanned with 38 kV X-rays at 196
mA and a nominal resolution of approximately 20 $\mu$m as the sample was rotated 180° in 0.6° steps with an exposure time of 300-350 ms. X-rays were detected using a 5 Mp (2240 x 2240) flat panel sensor utilizing 2 x 2 binning, and projection radiographs were averaged over four frames. The resulting 1120 x 1120 pixel radiographs were then reconstructed into 2D




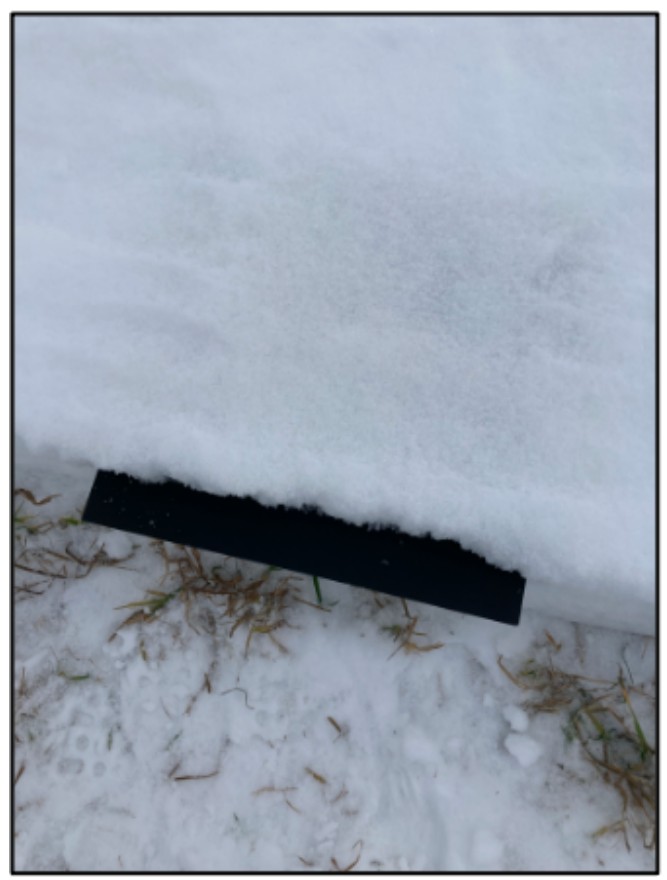

**Figure 5.** Photograph of the black aluminum panel inserted into the snow pit sidewall approximately 2.5 cm from the surface.

gray-scale horizontal slices using NRecon software (Bruker), which utilizes a modified Feldkamp cone-beam algorithm to produce a vertical stack of gray-scale cross-section images. Image reconstruction processing included sample-specific post
alignment, Gaussian smoothing using a kernel size of 2 to reduce noise, sample-specific ring artifact correction of dead pixels, beam hardening correction, and X-ray source thermal drift correction. A cylindrical volume of interest with a diameter of 1.6 cm was selected from the scanned samples in order to eliminate edge effects caused by the sampling process.

Resulting grayscale images are segmented into two phases: air (lowest X-ray absorption), and snow (highest X-ray absorption). Segmenting thresholds for each phase are determined by finding the local minimum between peaks on the histogram
showing all grayscale values, and using that value as a global threshold for each scanned sample. The resulting binarized data are despeckled so that any objects less than 2 pixels in diameter were removed.

The final binarized images are then used to construct 3D representations of dry snow samples for input into the RTM. This is accomplished through the use of open-source image processing and 3D visualization software packages accessed through Python (Schroeder et al., 2004; Van der Walt et al., 2014; Sullivan and Kaszynski, 2019). From the binarized images, the



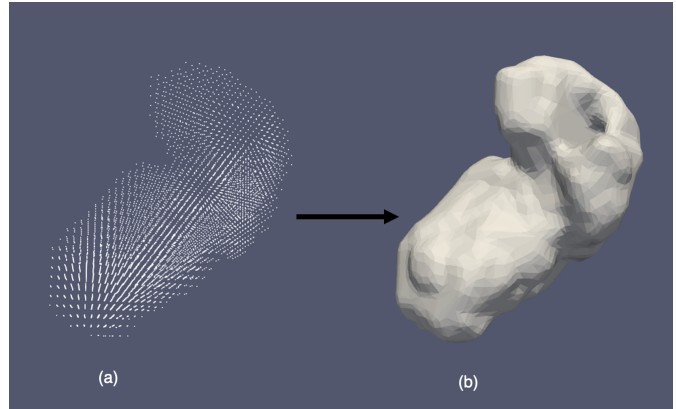

**Figure 6.** a) Point cloud representation of an example grain. b) surface rendering of the point cloud.

first step is to detect and mark individual snow grains. This can prove difficult as the grains are sintered together and their definition can be subjective. However, there is a wide body of literature on grain segmentation methods (e.g., Mangan and Whitaker, 1999; Wang et al., 2012; Theile and Schneebeli, 2011; Hagenmuller et al., 2013, 2014). Here, we use a common grain segmentation threshold technique called watershed segmentation due to its relatively easy implementation, computational efficiency, and accuracy. To apply this method we first combine a stack of binarized images and construct a 3D point cloud sub

sample of approximately 800 mm$^3$ ($\approx 450^3$ voxels) and calculate the Euclidean distance transform of the point cloud. Markers are then generated for the local maxima of the distance to the background. The watershed algorithm treats pixel values as a local topography and floods basins from the markers until basins attributed to different markers meet on watershed lines, which tend to occur along physical grain boundaries. From this, individual grains are extracted from the sample.

        To build a full sample mesh, the grains are processed on an individual basis and then combined together to create the

full mesh. To process each individual grain, a contour-based surface reconstruction process was developed to generate grain surfaces from the voxels that make up the grain. This method uses a subset of the binary sample array that contains the target grain, including both snow and adjacent air voxels. The subset array is then refined to increase the resolution. A Gaussian filter is applied to smooth the refined array, diminishing pixelated appearance of the voxelized snow-air interface, producing a smooth level set from which to extract the grain surface (Fig. 6). The smoothed level set is then used to define an isosurface at

the snow-air boundary, providing control over where the boundary is drawn with respect to the voxels.

        Finally, to extract the isosurface from the 3D voxel array, we apply the Marching Cubes method. In this technique, the input volume is divided into a discrete set of cubes. The algorithm then determines how the surface intersects with a given cube, based on the classification of the surrounding vertices and calculates an index for the cube by comparing the sample values at the vertices with the given isosurface value. From these results, it uses a pre-calculated lookup table of various surface-edge

intersections possible with the cube. Finally, the algorithm finds the surface-edge intersection through linear interpolation, resulting in a triangulated mesh. The algorithm "marches" through each cube. The original algorithm presented by Lorensen and Cline (1987) can lead to cracks and over the years has been improved by many (Nielson and Hamann, 1991; Scopigno,



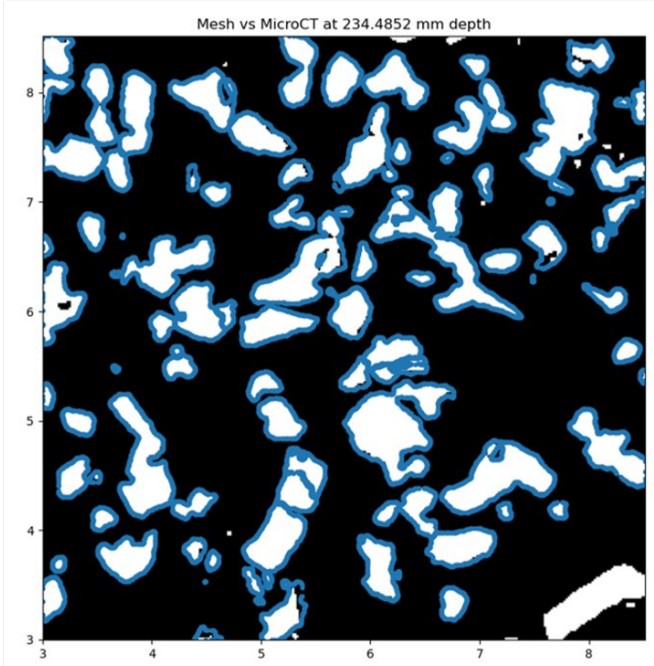

**Figure 7.** 2D Cross-sectional slice of a binarized $\mu$CT scan with corresponding mesh boundaries superimposed shown as the blue lines.

1994; Natarajan, 1994; Chernyaev, 1995; Lewiner et al., 2003). For this work, we used the adaptation implemented by Lewiner et al. (2003), which improved the algorithm to resolve face and internal ambiguities, extended the lookup table, and guaranteed

correct topology. As a final step, each grain is "repaired" to remove any defects and degenerate elements and ensure a manifold surface according to Attene (2010), and then decimated to reduce the overall number of triangles that comprise the surface thereby lowering the computational requirements. Overall, this method appears to accurately characterize the snow within the $\mu$CT sample with computed mesh snow sample densities within 1.5% of snow densities computed from the raw voxels. Figure 7 shows a 2D cross section comparing grain boundaries to the raw pixels of the image and selected example 3D rendered grains

are show in Figure 8.

## 3 Results

### 3.1 General Evaluation

An initial evaluation of the model is performed by simulating the spectral albedo for two idealized 20 cm deep snowpacks with uniform optical properties throughout. For these snowpacks, the optical properties are determined from 3D meshes generated

by two characteristically distinct $\mu$CT samples. One mesh is representative of fresh, fine-grained snow near the surface, and the other of large facets near the bottom of the snowpack (Fig. 9). For each mesh, the total mesh volume is approximately





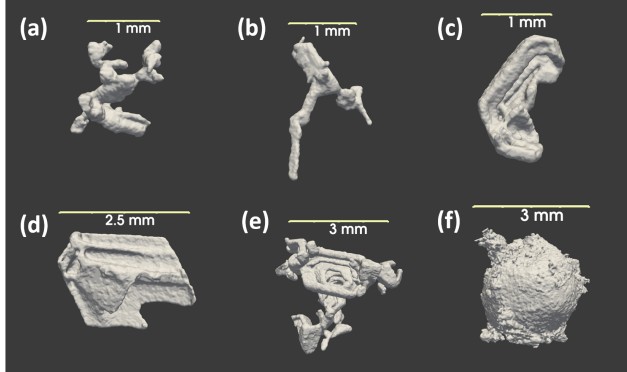

**Figure 8.** Example 3D renderings of different snow grains. These grains represent a variety of snow types including fresh snow aggregates (a, c), elongated needles (b), and faceted grain fragments (d,e), and sleet (f).

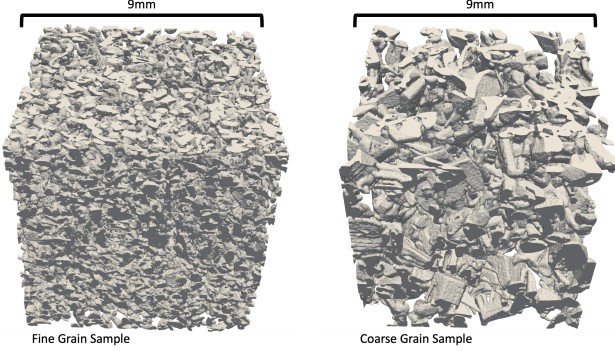

**Figure 9.** 3D renderings of mesh samples used to generate the optical properties for the general evaluation and snow transmissivity comparisons.

800 mm$^3$. Additional physical and optical properties of the each mesh are presented in Table 1. For each sample, the spectral albedo is computed for wavelengths between 400-1300 nm at 25 nm intervals with diffuse incident radiation. This comparison demonstrates that the model capably reproduces some known behavior of spectral albedo, and specifically, it shows a strong

320 wavelength dependence on snow microstructure that favors the NIR (Fig. 10a). The spectral albedo is relatively uniform between the two snowpacks for the spectral range between 400 and 800 nm, and then the albedos diverge, with a more rapid decrease in albedo for the coarser-grained snow.

  We then assess the relationship between simulated spectral albedo and incident zenith angle for the fine grain snow sample at four different wavelengths to evaluate the model's ability to simulate anisotropy in the surface reflectance (Fig. 10b). This

325 analysis shows an exponential increase in albedo at high zenith angles that is most pronounced in the NIR, consistent with observed behavior. This result indicates that the model is capable of simulating surface anisotropy with good fidelity. As a





**Table 1.** Physical and optical properties of the fine grain and coarse grain mesh samples. Note that SSA and $\rho_s$ are computed directly from the $\mu$CT sample.

| Property | Fine Grain | Coarse Grain |
|---|---|---|
| SSA (m$^2$ kg$^{-1}$) | 19.29 | 12.85 |
| $\rho_s$ (kg m$^{-3}$) | 263 | 232 |
| $\gamma_{ext}$ (mm$^{-1}$) | 1.86 | 0.99 |
| $F_{ice}$ | 0.44 | 0.41 |

related evaluation, the model-simulated DCRF is computed as a function of zenith angle (Fig.11). This analysis reveals that the reflectance is mostly isotropic for zenith angles less than approximately 55° at which point the surface becomes increasingly forward scattering, consistent with previous observational and modeling studies (Kaempfer et al., 2007; Dumont et al., 2010; Xiong et al., 2015; Jiao et al., 2019).

Finally, we use the model to provide an initial assessment of the impacts of snow microstructure on simulated spectral transmissivity. To accomplish this, the optical properties of the $\mu$CT samples in Fig. 9 are used to simulate and compare the spectral transmissivity at varying depths (Fig. 12). The transmissivity is highest at the short, non-absorptive, wavelengths and gradually decreases throughout the NIR, broadly matching quantitative snow transmissivity results reported in Perovich (2007) and Libois et al. (2013). The depth of the 5% transmissivity contour for the fine-grain snow sample is approximately 2.5 cm for the visible, and decreases to approximately 1.5 cm for the NIR (Fig. 12a), indicating that the fine-grain snow optical thickness is on the order of only a few centimeters. In contrast, the transmissivity for the coarse grain snow is increased near the surface, and the depth of the 5 % contour increases to 7.5 cm for the visible and 3 cm for the NIR (Fig. 12b).

### 3.2 Evaluation against UVD Data

The optical properties used in the 1D RTM were determined from four approximately 800 mm$^3$ $\mu$CT samples, with each sample representing a 2 cm thick layer within the top 8 cm of the snowpack. The RTM is then configured with 4 layers according to these optical properties (given in Table 2). The top three layers are each 2 cm thick, and the bottom layer is 28 cm thick, such that the entire snow depth amounted to 34 cm. We choose this configuration working under the hypothesis that the snow microstructure below 8 cm had little impact on the measured surface spectral albedo. To simulate the panels, the snowpack depth is modified to be 4.75 and 2.5 cm deep with a 100 % absorptive lower boundary while maintaining the layering corresponding to Table 2.

There is remarkably good agreement between our observations and the model (Fig. 13) and in particular, the model accurately simulates the impact of the inserted panel on the surface albedo for wavelengths shorter than 1000 nm for both the 4.75 and 2.5 cm depths. The model and observations diverge after 1200 nm, in particular, the simulated albedo is substantially higher in this range than measured. We hypothesize that this is primarily due to the phase function approximation computed from the rendered snow grains. Additional simulations that use the phase function computed from idealized spheres and columns support this



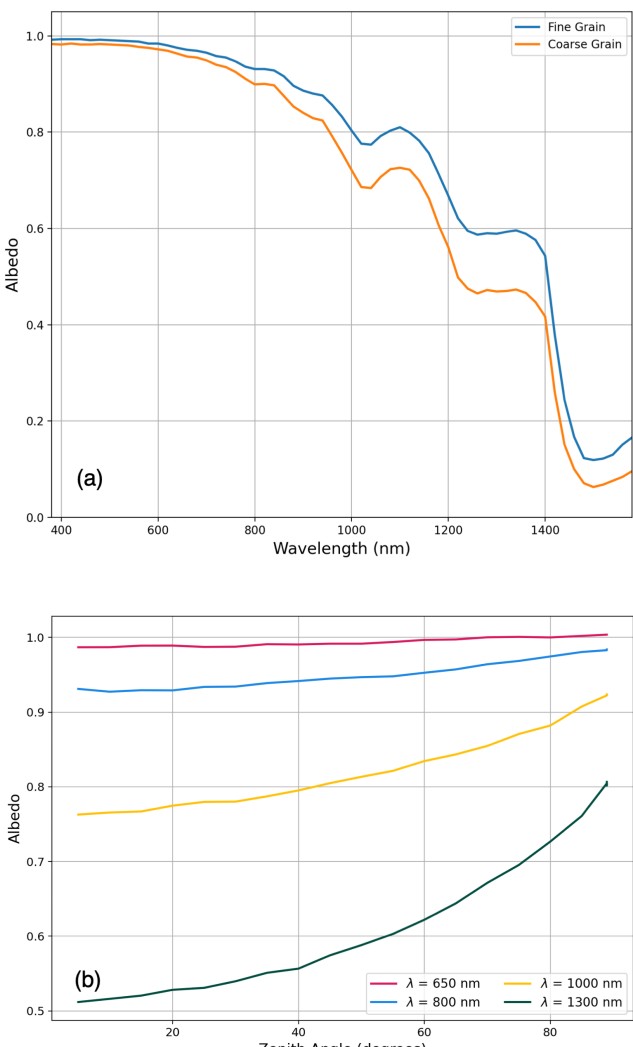

**Figure 10.** a) Simulated spectral albedo for fine grain and coarse grain snow samples for 100% diffuse radiation. b) Simulated spectral albedo as a function of incident zenith angle for selected wavelengths. Note that both simulations were run with 25000 photons.

hypothesis by showing that the spectral albedo is sensitive to the phase function at these wavelengths (not shown). This suggests that future improvements in the grain segmentation and surface rendering algorithms could improve these results in the NIR. Alternatively, this difference possibly illustrates the limits of the geometric optics approximation as the approximate particle





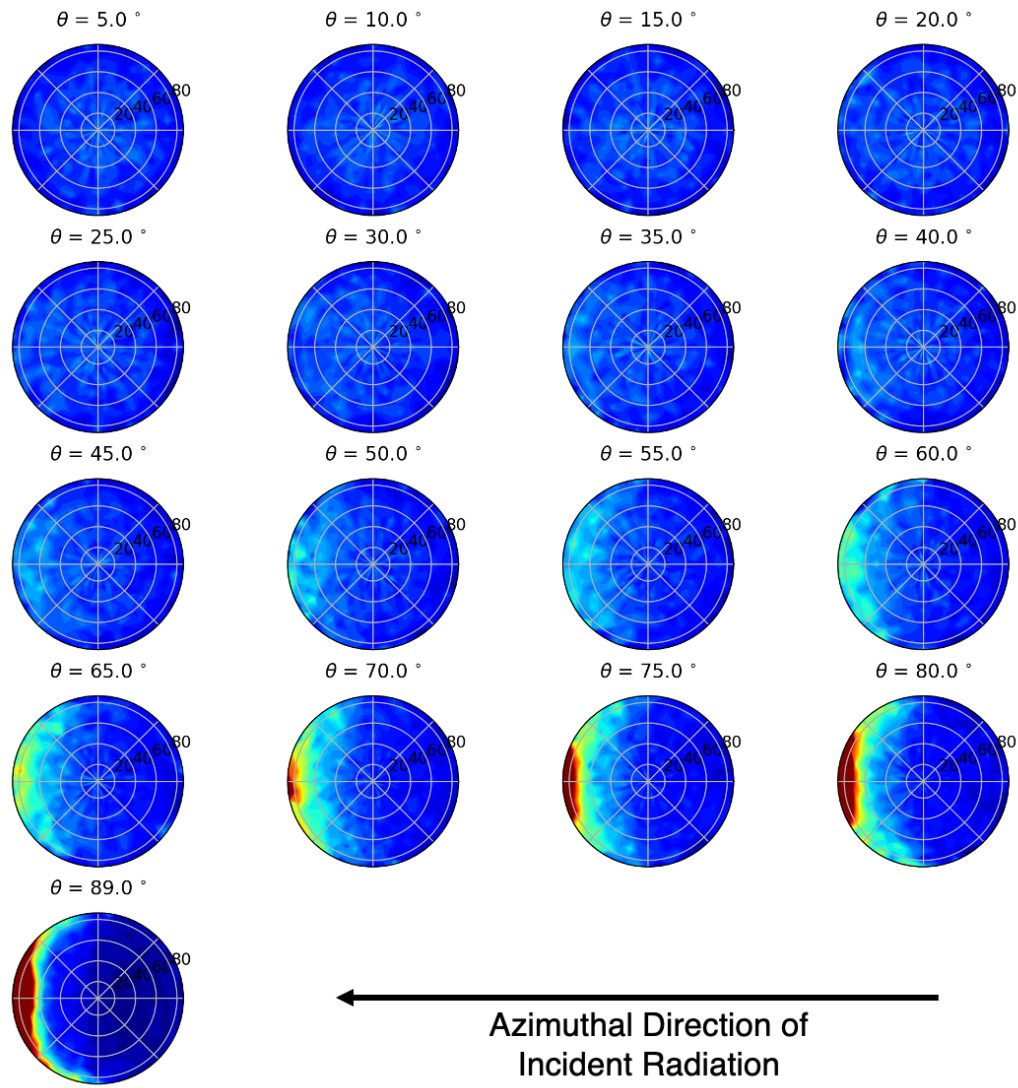

**Figure 11.** Polar plots of DCRF at 1000 nm for incident zenith angles ranging from 5 - 89°. Reflected azimuthal direction is on the theta axis, and reflected zenith angle if on the $r$ axis. Color scale ranges from 0-1.5.

355  size parameter is $< 1000$ for $\lambda > 1000$ nm. Overall, this initial evaluation against the ASD data is promising and suggests that this framework can be used to better understand the impacts of snow microstructure on reflectance and transmissivity.

### 3.3  Snow optical and physical properties

A final analysis is performed to determine how strongly common snow physical properties relate to the simulated optical properties from this framework. Specifically, we compare snow specific surface area (SSA) and snow sample density ($\rho_s$) to



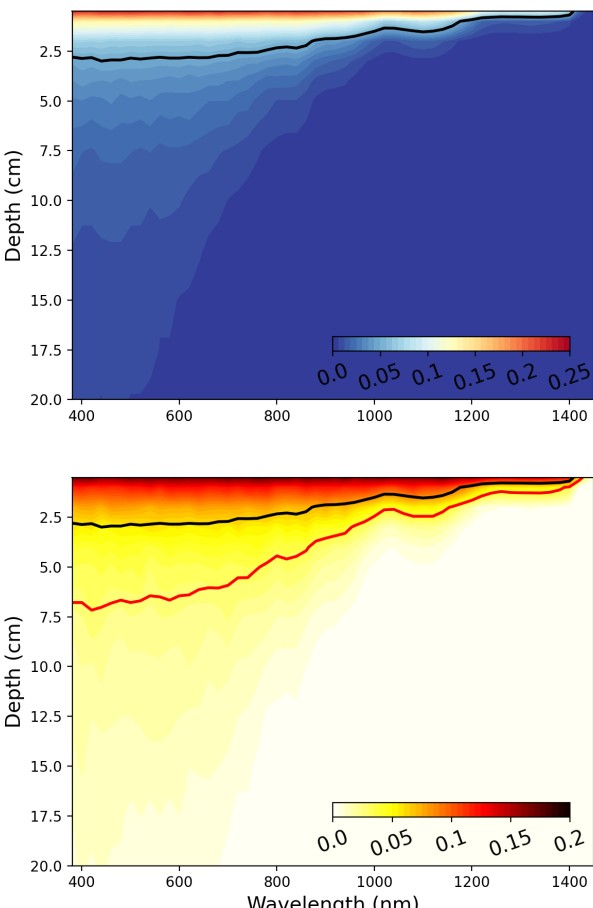

**Figure 12.** a) Simulated transmissivity of the fine grain snow sample contoured as a function of depth and wavelength. b) Transmissivity difference between the coarse grain and fine grain samples as a function of wavelength and depth. The 5% transmissivity contour is plotted for the fine grain (black) and coarse grain (red) samples.

**Table 2.** Physical and simulated optical properties of the top 8 cm of snow measured at the UVD site on February 12 2021. SSA and $\rho_s$ are computed directly from the $\mu$CT sample. Note that the depths correspond to the RTM model depths for the virgin snow calculation.

| depth [cm] | SSA (m$^2$ kg$^{-1}$) | $\rho_s$ (kg m$^{-3}$) | $\gamma_{ext}$ (mm$^{-1}$) | $F_{ice}$ |
|---|---|---|---|---|
| 1 (32-34) | 26.1 | 147 | 1.39 | 0.35 |
| 2 (30-32) | 27.2 | 178 | 1.77 | 0.38 |
| 3 (28-30) | 21.12 | 250 | 2.04 | 0.47 |
| 4 (0-28) | 18.44 | 287 | 1.86 | 0.51 |



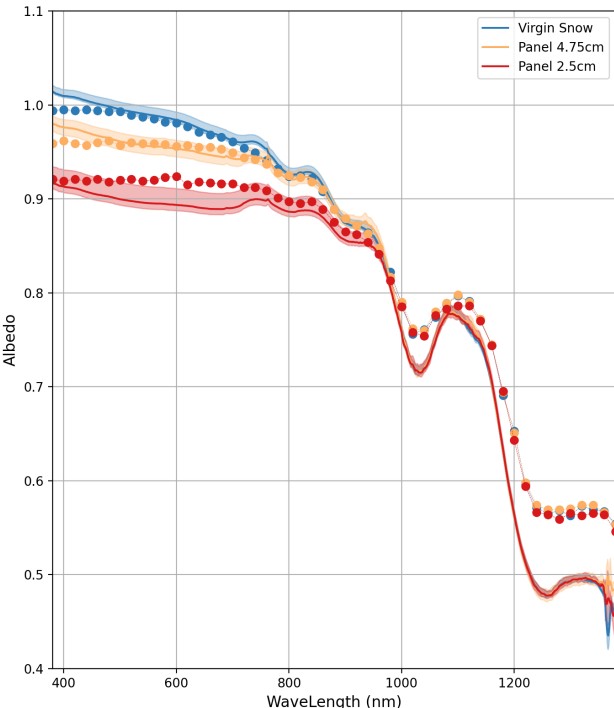

**Figure 13.** Simulated and observed spectral albedo at UVD for three different snow depths. Solid lines indicate observations and dotted lines indicate simulations. The shading around the observations indicates the inter-quartile range of the measurements computed from the five snow and two reference scans collected during each measurement, providing an assessment of measurement uncertainty. Mean RMSE of simulated albedo over 400-1600nm is equal to 0.06.

360    $\gamma_{ext}$ and $F_{ice}$. This analysis is performed using several $\mu$CT sample volumes collected on different dates, at different locations, and for various snow types. Note that each $\mu$CT sample is approximately 800 mm$^3$ and the sample SSA and $\rho_s$ are determined from the $\mu$CT 3D rendering. This analysis reveals that $F_{ice}$ has a very robust relationship with snow density (Fig. 14a) described by the linear fit:

$$F_{ice} = 0.00074\rho_s + 0.25, \qquad (21)$$

365      with $r^2 = 0.81$. In contrast, while $\gamma_{ext}$ generally seems to increase as a function of SSA, this relationship is not as well constrained as the relationship between $\rho_s$ and $F_{ice}$ (Fig. 14b). Instead, we find that $\gamma_{ext}$ is generally better approximated by a multivariate regression function that includes both SSA and density:

$$\gamma_{ext} = 0.077SSA + 0.0028\rho_s - 0.59, \qquad (22)$$





with $r^2 = 0.26$. Interestingly, this relationship can be improved substantially by removing the ice-crust samples from the dataset. This change modifies eq. 22 to:

$$\gamma_{ext} = 0.085 SSA + 0.0062\rho_s - 1.51, \tag{23}$$

and increases $r^2$ to from 0.25 to 0.79. This relationship dictates that the extinction coefficient increases both as a function of SSA and $\rho_s$, indicating that the highest extinction coefficients will be associated with small, tightly packed snow grains. This is qualitatively consistent, though not directly comparable, with analytical formulations in Kokhanovsky and Zege (2004), who show the extinction coefficient is related to both surface area and grain concentration, which is a good proxy for snow density. In contrast, with ice-crusts the extinction coefficient is more strongly related to SSA, and largely independent of $\rho_s$, leading to low extinction coefficients despite very high snow densities, counteracting the relationships described in equations 22,23. This behavior can be attributed to the fact that for crust layers, snow density is no longer a good proxy for particle concentration and therefore does not significantly affect the extinction coefficient. Overall, these results suggest that commonly observed physical snow properties can be used to approximate optical properties in conjunction with an appropriate phase function for the medium photon-tracking RTM model for non-crust snow layers.

To further assess how these two specific snow optical properties, $\gamma_{ext}$ and $F_{ice}$, affect the greater simulated spectral transmissivity, we perform a sensitivity analysis by comparing the 5% transmissivity contour depth for three fractional ice paths: 0.30, 0.46, 0.69 at two fixed $\gamma_{ext}$ values: 2.34, 0.81 mm$^{-1}$. The two $\gamma_{ext}$ values correspond to the max, min values found in the previous analysis and presented in Fig. 14b. The three $F_{ice}$ values correspond to the max, min, and mean values (Fig. 14a). We compare the influence of $F_{ice}$ at both the max and min $\gamma_{ext}$ values, since we anticipate the strength of its influence will vary according to $\gamma_{ext}$. While we pair the maximum $\gamma_{ext}$ with the $F_{ice}$, we note that high values of $\gamma_{ext}$ are more likely to coincide with high values of $F_{ice}$ due to the shared dependence of these variables on snow density in most snowpacks.

The results of this analysis indicate that $\gamma_{ext}$ has a much larger impact on snow transmissivity than $F_{ice}$. This is unsurprising due to the highly scattering nature of snow. In particular the snow medium is much more transmissive at the minimum $\gamma_{ext}$ value with the 5 % transmissivity contours exceeding 7 cm in the visible and 3 cm in the NIR. This is in contrast to the transmissivity of the maximum extinction coefficient, which never exceeds 4 cm of depth (Fig. 15). $F_{ice}$ can either amplify or dampen the effect of $\gamma_{ext}$ in the NIR, with lower values of $F_{ice}$ leading to in increase in transmissivity at a given depth compared to high values of $F_{ice}$.

## 4   Discussion

One key objective of this work is to expand beyond determining snow optical properties from a specified distribution of grains with idealized shapes, and instead represent snow as an organized structure in determining them. One snow type where this approach may be particularly advantageous is in understanding the optical properties of snow crust layers, which do not fit easily into the collection of particles or air bubbles approximation. This is supported by our finding that $\gamma_{ext}$ is very well



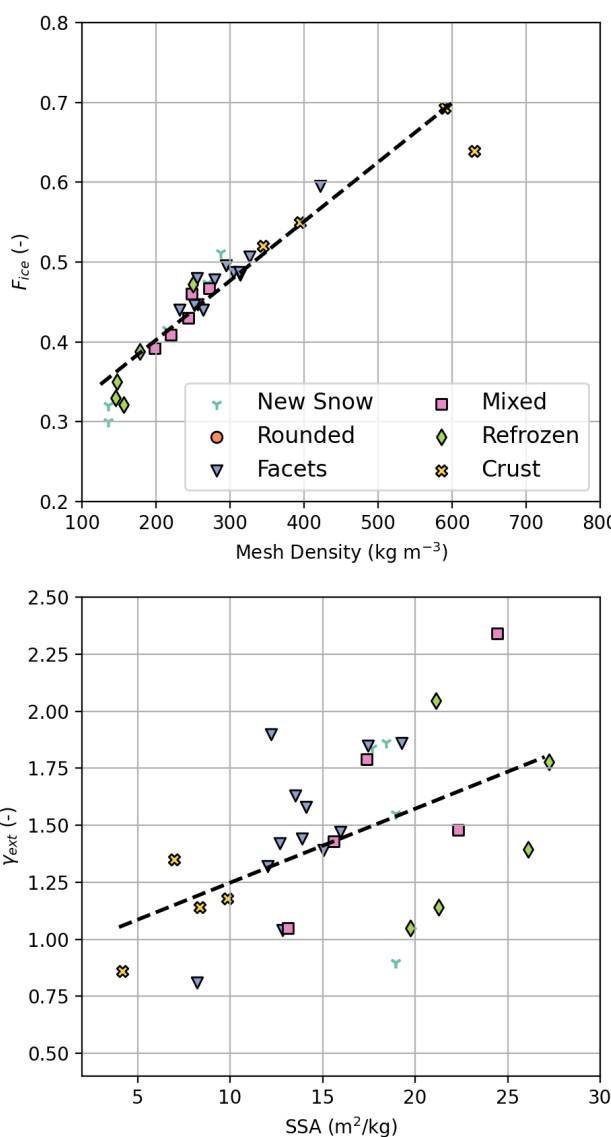

**Figure 14.** Optical properties for $\lambda$=900nm computed from $\mu$CT photon-tracking compared against sample physical properties. a) $F_{ice}$ vs. $\rho_s$ and b) $\gamma_{ext}$ vs. SSA. Linear regression lines are shown in black dashed lines. Note that the regression line plotted on (b) is a single-parameter regression line for illustrative purposes, and not the multi-parameter regression discussed in the text. Snow grain forms were determined through visual assessment during snow pit analysis.

approximated with a multivariate regression of SSA and $\rho_s$ for all observed snow types except for crusts, which exhibited the



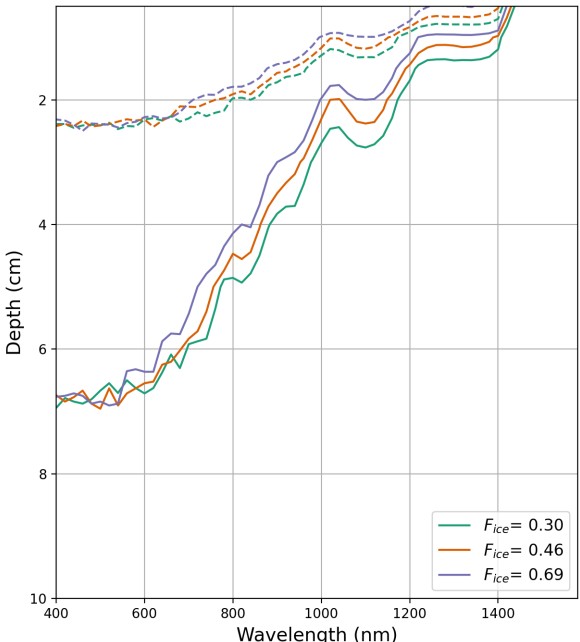

**Figure 15.** Depth of simulated 5% transmissivity contour as as a function of wavelength for varying $F_{ice}$ at two extinction coefficient: $\gamma_{ext} = 0.81$ (Solid lines) and $\gamma_{ext} = 2.34$ (Dashed lines).

highest snow densities and the lowest extinction coefficients and SSAs. Another snow type this approach may be well suited for is highly aged snow that has metamorphosed into a collection of large irregularly shaped grain clusters and pore spaces.

While ideally, the optical properties would vary slightly with wavelength due to the impact of absorption on $\gamma_{ext}$ and the phase function, we chose to leave the optical properties independent of wavelength. This choice is made entirely to reduce
the computational burden of running the photon-tracking model for several wavelengths. Cursory sensitivity tests performed to assess the impact of this choice on the optical properties supported the use of wavelength-independent optical properties, as both $p(cos\Theta)$ and $\gamma_{ext}$ exhibited generally a negligible dependence on wavelength for $\lambda > 1400$. We suspect that this is due to the fact that a ray is much more likely to be scattered at an air/ice boundary than by absorption within the particle (i.e., snow has a high single scattering albedo). This may not be the case for all snow types, in particular for the NIR wavelengths within
very large grains, and is worthy of future exploration.





# 5 Conclusions

In this work we have presented a blended photon-tracking radiative transfer model in an effort to better understand the complicated influence of snowpack microstructure on snow spectral transmissivity in the geometric optics limit. A foundational goal of this modeling approach is to expand upon previous approaches aimed at incorporating 3D renderings of real snow
microstructure into radiative transfer models for snowpacks of arbitrary depth, while maintaining the Monte Carlo aspects of the model. To accomplish this, existing methods for simulated photon interactions with rendered elements are employed to determine key optical properties of the snow (Grundy et al., 2000; Kaempfer et al., 2007; Xiong et al., 2015).

An evaluation of this framework for consistency with known behavior of spectral snow albedo revealed that this framework can successfully reproduce the dependency of spectral albedo and grain size, as well as the surface anisotropy at high incident
zenith angles. Furthermore, a comparison of the simulated snow albedo against spectroradiometer measurements collected in the field over snow with varying depths indicate that the model can simulate the effects of an underlying surface on spectral albedo with high accuracy.

In comparing two different snow samples, it was revealed that snow microstructure has a large impact on snow transmissivity in the visible spectrum and near the snow surface, increasing the 5 % transmissivity depth from approximately 4 cm for a fine
grain snow sample to 7.5 cm depth for a coarse grain sample. A brief sensitivity analysis of the optical properties revealed that lowering the medium extinction coefficient acted to reduce the albedo and increase transmissivity in the visible bands, while the fractional ice path ($F_{ice}$) impacted the rate at which albedo and transmissivity decreased as a function of wavelength in the NIR. Accordingly, we anticipate that snowpacks made up of large grains with low fractional ice-paths will be the most transmissive.

Overall, while current efforts are focused on using this model to better understand snow transmissivity, it shows promise as a broadly applicable snow RTM that has a strong direct connection to $\mu$CT snow samples. While currently, it is limited to the geometric optics approximation for clean snow and unpolarized radiation, ongoing and anticipated future efforts are aimed at improving the grain segmentation and rendering process, incorporating polarization, parameterizing diffraction, and including light absorbing particulates (LAPs). In particular, recent multiphase image segmentation techniques (e.g., West et al., 2018)
could be used to better separate snow, air, and LAPs in a $\mu$CT sample allowing for the impact of LAPs to be determined through ray-tracing. Furthermore, because the model operates entirely as a photon-tracking model, it is a natural fit with macro-scale ray-tracing and therefore could be used to investigate the reflectance of rough snow surfaces such as sun cups or sastrugi.

*Code and data availability.* The mesh generation and RTM code with associated documentation is available in preliminary 'as is' format on Github at (lhttps://github.com/wxted/CRREL-GOSRT.git). Sample data files used to generate figure 9a is available on Github as sample
data. Additional limited sample data, including rendered microCT meshes, and spectroradiometer data used for this paper are available upon request



*Author contributions.* Theodore Letcher performed a majority of the model physics, structural development, and coding, in addition to coordinating the model analysis and manuscript preparation. Julie Parno led research and coding efforts related to the 3D mesh generation and rendering and assisted in general coding, she also coordinated a majority of the fieldwork activity. Zoe Courville provided research support,

participated in snow sampling and coordinated $\mu$CT analysis. Lauren Farnsworth performed a large portion of $\mu$CT scans and a majority of the $\mu$CT image post processing and analysis. Jason Olivier participated in fieldwork and provided background on the ASD instrumentation and sampling for the manuscript. Theodore, Julie, and Jason performed the RTM simulations and assisted in code debugging. All authors provided writing support for the manuscript.

*Competing interests.* The authors declare that no competing interests are present

*Acknowledgements.* We wish to acknowledge Dr. Arnold Song from Dartmouth College in Hanover New Hampshire for providing relevant code examples to facilitate 3D shape rendering. We also wish to thank Taylor Hodgdon from CRREL for providing occasional support and advice on using Python to perform snow grain segmentation and rendering. Finally, we wish to thank Dr. Bert Davis, and Dr. Ned Bair for providing valuable initial feedback on the manuscript. Funding support for this research was provided by the U.S. Army Engineer Research and Development Center (ERDC) Cold Weather Military Research project. **Place Holder to thank reviewers**



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
