# Peer review of "A generalized photon-tracking approach to simulate spectral snow albedo and transmissivity using X-ray microtomography and geometric optics"

_The Cryosphere, 2021_

## Referee Comment (RC1)

Review of « A generalized photon-tracking approach to simulate spectral snow albedo and transmissivity using X-ray microtomography and geometric optics », by Theodore Lechter et al.

**General comments**

This paper describes a novel approach to extract relevant snow optical properties, namely the mean fractional ice path, extinction coefficient and phase function, from X-ray microtomography samples. Using these optical properties, a plane-parallel Monte-Carlo radiative transfer model is run to simulate the reflectance and transmittance of a multi-layer snowpack. This blended model is first applied to a single layer ideal snowpack, and then to a real, layered snowpack that was probed on 12 February 2021 at Union Village Dam in Thetford, Vermont. For this snowpack, along with the snow samples, spectral reflectance measurements were performed above the snowpack, including mesurements with a black panel inserted at 2.5 and 4.75 cm below the surface. This model reproduces the expected sensitivity of transmittance and reflectance to snow SSA, as well as the strong anisotropy of snow reflectance. The comparison between simulated and observed reflectance is satisfactory, although large discrepancies remain in the near-infrared, which are attributed to deficiencies in the estimation of the phase function.

The paper is well written and the figures are clear. The topic perfectly fits in the scope of The Cryosphere, and estimating local optical properties from X-ray microtomography snow samples is certainly a question that deserves more research. The development of a new approach and its tentative validation with in situ measurements is a great contribution to tackling this question. In that sense this paper contains sufficient original material to be considered for publication. However, several critical issues remain, mostly regarding the estimation of snow optical properties, that preclude accepting the paper before major revisions are performed. The experimental validation step is not very convincing yet, and the discussion does not sufficiently dwell on the obvious limitations of the present study. As a result, the reader is let with poor confidence that the proposed model can reliably simulate snow optical properties, including transmittance which is the focus of the paper but is not supported by any experimental validation.

**Specific comments**

1) The phase function of the snow is estimated by first isolating individual snow grains, and using a ray-tracing code to estimate their single-scattering properties. Although there is probably no easy way to estimate the local phase function of a bicontinuous medium, this approach is very questionable. First there is no evidence that a snow sample can be separated into individual grains without making very arbitrary choices. Think for instance of old snow that resembles more a porous medium than an ancient collection of individual snow particles. Second, the number of such grains to be selected is not discussed while it is clearly a limitation of the approach. The approach proposed by Xiong et al. (2015) to estimate the phase function of a bicontinuous medium is more aligned with the strategy used to estimate the extinction coefficient, and could probably at least be compared to the current approach. Approaches used in other disciplines (scattering in any porous medium) may also provide interesting alternatives (see for instance Haussener et al., 2012). In addition, it would be great to provide the values of the asymmetry parameter computed for the estimated phase functions, which would allow comparison with usual assumptions made on this hardly measurable quantity.

2) A method is proposed to estimate the extinction coefficient of snow from the 3D images, which is based on sampling the probability to be scattered or absorbed when traveling a certain distance from a variety of locations chosen randomly. This strategy seems acceptable, but does not reproduce the dependence of $\gamma_{ext}$ on SSA and density $\rho$ expected for a collection of individual convex particles, that is $\gamma_{ext} = \rho \cdot SSA / 2$. This simple relation has been used in recent, similar, studies, though

(Malinka, 2014, Picard et al., 2016). Although this deviation from the expected behavior is already seen in the original paper by Xiong et al. (2015) (their figure 4c), it should be further investigated. For instance it would be worth checking that the present strategy reproduces the expected dependence in the case of a collection of spheres. If it were carefully confirmed that in real snow the relation $\gamma_{ext} = \rho \cdot SSA / 2$ does not hold, this would be an important result. Not also that Figure 14 should be recomputed, looking for a correlation between $\gamma_{ext}$ and the product $\rho \cdot SSA$.

3) To my knowledge, the use of the mean fractional ice path $F_{ice}$ is quite unique to this study. Generally, the medium is instead represented by its absorption efficiency, or equivalently by its single-scattering albedo. In Eq.18, the energy of the photon packet is decreased along the path *s* due to absorption within the ice. However the distance traveled in the snow is underestimated when using a continuous representation, because the distance $l_{ext}$ between scattering (or absorption) events is less than the actual distance traveled, which should include internal reflections not accounted for in $l_{ext}$ (see Malinka 2014). As a consequence I believe (this has to be verified) absorption is overall underestimated with the present method. This can be easily corrected by carefully taking into account internal reflections (as is made for the estimation of $F_{ice}$). Hence I'd encourage the authors to compare their approach to a more standard one based on single-scattering albedo (in which case absorption is seen as a probabilistic localized event rather than a continuous process). This would help validate the $F_{ice}$ approach or point to fundamental and critical differences between both approaches. Also, it should be physically explained why a linear relation is expected between $\rho$ and $F_{ice}$.

4) Related to the previous point, I stress that some work has been focused in the past decade on the estimation of the absorption enhancement parameter B of snow (Libois et al., 2013, 2014, 2019), which is actually directly related to the proposed definition of $F_{ice}$. Estimations of B from the joint values of $F_{ice}$ and $\rho$ provide results quite different from those obtained elsewhere, which at least deserves comments, if it does not help pointing to deficiencies of the present study. Although this could in itself form relevant material for an independent, complementary study, it appears necessary to at least clarify this point.

5) The paper tries to differentiate from previous similar studies by focusing on transmissivity, in complement to more widely explored reflectance properties. However, only a few observations (practically, 3 reflectance spectra) are used to validate the model, which do not correspond to the primary focus of the work. Since these measurements are not very well simulated (compared for instance to those reported by Carmagnola et al., 2013), it does not give confidence in the transmissivity simulations, in particular in the near-infrared. We also note than in the visible some differences remain that could be partly explained by the presence of light-absorbing impurities, which is not really discussed.

6) One of the main problems of this paper is that it lacks a proper, critical discussion. In the current version the discussion is 15 lines long and does not really question the whole results of the paper. I'm convinced that given the uncertainties arising from the chosen method to estimate local optical properties, the issues related to the spatial representativity of the very small samples, and the very limited number of observations that unsuccessfully try to support the model, a much longer discussion would be very useful.

**Technical corrections**

l.3 : "based on X-ray microtomography" is unclear → "reflectivity of snow samples based on X-ray microtomography images ?

l.6 : is really the focus more on transmissivity than albedo?

l.7 : sub-nivean hazard detection is mentioned in the abstract but not later on

l.8 : should snow grain size be replaced by SSA? Because the advantage of having 3D images of snow samples is to get rid of the simple, unrealistic, granular approach

l.10 : not clear whether in the field transmissivity and/or reflectivity measurements were performed

l.13 : "is limited to the top 5 cm" and "can penetrate" is awkard. Transmissivity is a property. Should be rephrased.

l.14 – 15 : I think this result is quite obvious. Maybe consider providing another more specific result.

l.28 : "aggregate" is unclear

l.42 : to state that geometric optics works well some hint should be given about the typical size of scatterers in snow (or referring to a paper stating why it works well)

l.43 : I think the interaction between a "snow particle" and light is far from being well understood. Because we essentially don't know what a snow particle is. Consider rephrasing the sentence.

l.47 : Here Mie theory is mentioned which seems to contradict the fact that geometric optics is used. Outside of the Mie regime, using a Mie code is probably useless.

l.49 : I'd say that Mie theory CAN ONLY BE (by definition) applied to spherical particles

l.53 : please double-check that the errors highlighted by Dang et al. (2019) indeed result from the spherical hypothesis, and not from the two-stream approximation

l.59 : Is it similar to the approach of Malinka (2014), which could then be cited as well?

l.62 : I'm not sure in Dumont et al. (2021) they used ray-tracing on numerical samples to simulate optical properties.

l.65 : I'd encourage the authors to use reflectance/transmittance (the measured quantity) or transmissivity/reflectivity (the material property), but not a mixture of both.

l.72 : I understand that the main difference of this study with the previous ones upon which it builds is the special focus on transmissivity. If it's the case, consider being even more specific on this point.

l.75 : what does "semi-quantized" mean?

l.89 : "RT through non-spherical properties" is unclear. Do you mean to simulate single scattering properties of such particles?

l.83 : again, is the reference to Dumont et al. (2021) appropriate?

l.90 : remove "medium"

l.92 : how does this resolution compare with previous studies? Is it estimated to be sufficient to represent small-scale snow features that can have an impact on snow optical properties?

l.92 : does "1D" mean plane-parallel, that is horizontally homogeneous layers? Maybe clarify this

l.99 : Doesn't the 1D model need single-scattering albedo $\omega$? Would it be more appropriate to introduce first the 1D model (reverse 2.1 and 2.2) to highlight what properties are needed? Maybe consider to make the link between $F_{ice}$ and $\omega$ as follows (valid for weakly absorbing media):

$F_{ice} = B \cdot \rho/(\rho_{ice}+(B-1)\rho)$, where B is the absorption enhancement parameter (see Libois et al., 2019) And $1- \omega = 2B\kappa/(\rho_{ice} SSA)$ (Picard et al., 2016)

l.102 : unclear. Should it be the distance between extinction events (which can either be scattering or absorption)? And the relation is simply the inverse, no? Could you use there the more straighforward expression of extinction coefficient as density x SSA / 2 (depending on whether you include diffraction or not in scattering). See for instance Malinka (2014).

l.115 : could you comment on this wavelength dependence. Is it a default of the method, or something expected?

Eq. (4) : typo sign error; also, sin and cos should not be italic (same think throughout the paper)

l.137 : add $v_r$ and $v_t$ in parentheses here, and remove the end of the sentence l.141

l. 142 : this sounds very awkward to mention individual snow grains here, while the advantage of working with X-ray microtomogography images is to get rid of the particular representation of snow. Also, it is obvious that isolating grains from such an image is very arbitrary and isolated grains can behave very differently than the same grains being slightly sintered. Definitely it's not tricky to define the phase function of a porous medium based on local characterization (see for instance Haussener et al., 2012). This deserves more caution. The approach of Xiong et al. (2015) might be more appropriate than isolating "snow grains". An illustration of the segmentation process would be useful if this strategy is maintained. Could you also provide (where it best suits) the values of the asymmetry parameters obtained with this approach?

l.170 : this grain selection seems very arbitrary and would deserve more attention or explanations. Also, how many grains should be averaged to have something representative? What's the variability of the phase function across grains from a same sample?

l.172 : some comment is needed on the relevance of using the properties of a very small sample to represent a whole (necessarily heterogeneous) snowpack. Said differently, what is the representativity of the sample?

l.175 : reference to Picard et al. (2016) might be relevant

l.176 : problem with the beginning of the sentence

l.198 : could you double-check the equation for $\mu_y$. There might be a sign error. Also didn't you forget the last terms for $\mu_x$ and $\mu_y$?

l.202 : is this approach the initiative of the authors, or was it taken from another paper? I'm afraid it's wrong because it overall underestimates the total distance traveled by a photon (the distance

traveled between scattering events does not include enhancement in the ice phase), hence the total absorption. This is probably tricky.

l.207 : the "Russian roulette" should probably be better explained. What happens to the photon packets that are not killed?

Figure 3 : labels should be larger (also for Figs. 9, 10, 13). Why using markers? Why not simply having two layer with different shades?

l.227 : I did not understand "is used to simplify the complex reflectance properties of a rough surface"

l.238 : not clear whether samples are taken from the surface or in the pit (to sample various layers)

l.241 : how later (compared to snow sampling) were the images taken?

l.245 : "optimization" is unclear

l.247 : was it nadir observations?

l.255 : what's the size of the aluminium panel? How was it practically inserted? How did you ensure that it is horizontal under the snow? What's the precision of the depth position (transmissivity greatly varies with depth)? 4.75 cm seems very (too) accurate for a depth measurement! Typo for "aluminium"

Figure 4 : maybe not very useful.

Figure 5 : An indicative scale would be helpful

l.271 : Is the cylinder still 7 cm high for the detailed analysis, or limited to a cubic sample?

l.281 : totally agree, and this should be further discussed. Does this literature correspond to optical studies? If not, I see no reason that this approach is satisfying for optical issues.

l.313 : what is the underlying albedo? Visible light can indeed reach the ground

l.317 : "of the each"

l.318 : how is albedo computed? What are the illuminating and viewing angles?

l.320 : "that favors" reads awkward → "it shows the strong sensitivity of NIR albedo to snow microstructure"

l.323 : it's more a dependence than a relationship

l.325 : is exponential qualitative or confirmed? Consider referring to Eq. 9 of Kokhanovsky and Zege (2004)

Figure 10 : 25000 photons per wavelength (how many by the way?) or for all wavelengths?

l.326 : what "observed" behavior? Your work or from the literature?

l.326 : "with good fidelity" is not justified. What do you compare to? More generally your model could not but reproduce that, so it's not a proof of the model being "good".

Table 1 : would be useful to provide the parameter B as well (see Libois et al., 2014). In particular the values would be 1.95 for fine grain and 2.05 for coarse grain (TBC). Also $\gamma_{ext}$ is quite different from $\rho \cdot SSA/2$ (see Libois et al. 2013 or Picard et al., 2016), which would certainly deserve some comment. Have you applied your technique to a collection of isolated spheres (with known $\rho$ and SSA) to check whether you can obtain the exact (known) value of $\gamma_{ext}$?

l.332 : it's not clear what transmissivity is here. It it the flux at a particular depth within the snow layer of 20 cm? Or it it the transmittance of a snow layer of thickness X (with black surface beneath), where X is varying. Without precision I'd assume it's option 2. This makes a big difference and should be clarified.

l.336 : an optical thickness is unitless. The penetration depth is a length, though

Figure 12b: shouldn't the lines be removed, and the red line added to 12a? Because they correspond to transmissivities, not differences in transmissivities

Table 2 : as for Table 1, I found B values ranging from 2.3 to 2.8, which are somehow much larger than previous estimates.  Again $\gamma_{ext}$ is quite different (30%) from $\rho \cdot SSA/2$ (or even $\rho \cdot SSA/4$ if diffraction is not considered). Also, depth seems to start from the ground, which is surprising (looks more like height). Finally, I doubt that SSA estimation can reach 0.01 $m^2$ $kg^{-1}$ precision.

l.345 : why not using the 4% reflectance mentioned earlier?

l.347 : I don't understand what "remarkably good" means, given that substantial differences are seen in Fig. 13. Why are the differences in the NIR so large?

Figure 13 : why is there a line in between points?

l.350 : errors in the phase function are mostly impacting the NIR, but they also affect the visible range, which is why it deserves much more attention.

l.363 : according to the formula presented above ($F_{ice} = B \cdot \rho/(\rho_{ice}+(B-1)\rho)$) it's no surprise to have a linear-like behavior in the explored range of snow densities. However the 0.25 residual is surprising. I'd be worth computing B for all available data based on your approach.

l.372 : this has been known for a while (Eq. 23 of Bohren and Barkstrom, 1974)

l.368 : again, use the product $\rho \cdot SSA$ instead of a bilinear regression. Again a residual in such regression should be commented.

l.372 : what does a $r^2$ of 0.25 mean in terms of correlation?

l.380 : suggest → confirm

l.387 : can you clarify what "we pair" means

l.390 : not clear why the highly scattering nature of snow explains the different sensitivities to $F_{ice}$ and $\gamma_{ext}$. I'd use an analytical expression of the penetration depth (see for instance Libois et al., 2013) to show that the dependence on $\gamma_{ext}$ is linear, while the dependence on B is square root

(changes in $F_{ice}$ are proportional to changes in B if density is unchanged). The ~3 scaling of $\gamma_{ext}$ results in a ~3 scaling of penetration depth. The ~2.3 scaling in $F_{ice}$ results in a ~1.5 scaling, which is consistent with your simulations.

l.392 : never exceeds 2.5 cm, no?

l.393 : leading to **an** increase

l.398 : for crust layers, how would you isolate grains from the 3D image?

l.404 : at what wavelength were the phase function and $\gamma_{ext}$ estimated ?

l.407 : unit missing

l.422 : I don't agree with "high accuracy"

l.425 : at what wavelength?

l.426 : was the reduction of albedo mentioned earlier?

l.428 : again, this is obvious

l.435 : consider reading Hagenmuller et al. (2019)

*Additional suggestion* :

Here is a procedure to test the validity of the 1D ray-tracing code (although I don't particular believe it does not work):

- Consider an homogenous (horizontally semi-infinite) layer of thickness L, non absorbing
- Illuminate it with diffuse light
- Record the path lengths of escaping (reflected and transmitted) photons
- Check that the average path length equals 2L, **whatever the chosen phase function**
- If it does not work there is an issue somewhere
- See Blanco and Fournier (2003) for more details

**References**

Blanco, S., & Fournier, R. (2003). An invariance property of diffusive random walks. *EPL (Europhysics Letters)*, *61*(2), 168.

Bohren, C. F., & Barkstrom, B. R. (1974). Theory of the optical properties of snow. *Journal of Geophysical Research*, *79*(30), 4527-4535.

Carmagnola, C. M., Domine, F., Dumont, M., Wright, P., Strellis, B., Bergin, M., ... & Morin, S. (2013). Snow spectral albedo at Summit, Greenland: measurements and numerical simulations based on physical and chemical properties of the snowpack. *The Cryosphere*, *7*(4), 1139-1160.

Hagenmuller, P., Flin, F., Dumont, M., Tuzet, F., Peinke, I., Lapalus, P., ... & Charrier, P. (2019). Motion of dust particles in dry snow under temperature gradient metamorphism. *The Cryosphere*, *13*(9), 2345-2359.

Haussener, S., Gergely, M., Schneebeli, M., & Steinfeld, A. (2012). Determination of the macroscopic optical properties of snow based on exact morphology and direct pore-level heat transfer modeling. *Journal of Geophysical Research: Earth Surface*, *117*(F3).

Kokhanovsky, A. A., & Zege, E. P. (2004). Scattering optics of snow. *Applied Optics*, *43*(7), 1589-1602.

Libois, Q., Lévesque-Desrosiers, F., Lambert-Girard, S., Thibault, S., & Domine, F. (2019). Optical porosimetry of weakly absorbing porous materials. *Optics express*, *27*(16), 22983-22993.

Libois, Q., Picard, G., Dumont, M., Arnaud, L., Sergent, C., Pougatch, E., ... & Vial, D. (2014). Experimental determination of the absorption enhancement parameter of snow. *Journal of Glaciology*, *60*(222), 714-724.

Picard, G., Libois, Q., & Arnaud, L. (2016). Refinement of the ice absorption spectrum in the visible using radiance profile measurements in Antarctic snow. *The Cryosphere, 10*(6), 2655-2672.

---

## Author Comment (AC1)

3Review of « A generalized photon-tracking approach to simulate spectral snow albedo and transmissivity using X-ray microtomography and geometric optics », by Theodore Lechter et al.

**General comments**

This paper describes a novel approach to extract relevant snow optical properties, namely the mean fractional ice path, extinction coefficient and phase function, from X-ray microtomography samples. Using these optical properties, a plane-parallel Monte-Carlo radiative transfer model is run to simulate the reflectance and transmittance of a multi-layer snowpack. This blended model is first applied to a single layer ideal snowpack, and then to a real, layered snowpack that was probed on 12 February 2021 at Union Village Dam in Thetford, Vermont. For this snowpack, along with the snow samples, spectral reflectance measurements were performed above the snowpack, including mesurements with a black panel inserted at 2.5 and 4.75 cm below the surface. This model reproduces the expected sensitivity of transmittance and reflectance to snow SSA, as well as the strong anisotropy of snow reflectance. The comparison between simulated and observed reflectance is satisfactory, although large discrepancies remain in the near-infrared, which are attributed to deficiencies in the estimation of the phase function.

The paper is well written and the figures are clear. The topic perfectly fits in the scope of The Cryosphere, and estimating local optical properties from X-ray microtomography snow samples is certainly a question that deserves more research. The development of a new approach and its tentative validation with in situ measurements is a great contribution to tackling this question. In that sense this paper contains sufficient original material to be considered for publication. However, several critical issues remain, mostly regarding the estimation of snow optical properties, that preclude accepting the paper before major revisions are performed. The experimental validation step is not very convincing yet, and the discussion does not sufficiently dwell on the obvious limitations of the present study. As a result, the reader is let with poor confidence that the proposed model can reliably simulate snow optical properties, including transmittance which is the focus of the paper but is not supported by any experimental validation.

**Specific comments**

1) The phase function of the snow is estimated by first isolating individual snow grains, and using a ray-tracing code to estimate their single-scattering properties. Although there is probably no easy way to estimate the local phase function of a bicontinuous medium, this approach is very questionable. First there is no evidence that a snow sample can be separated into individual grains without making very arbitrary choices. Think for instance of old snow that resembles more a porous medium than an ancient collection of individual snow particles. Second, the number of such grainsto be selected is not discussed while it is clearly a limitation of the approach. The approach proposed by Xiong et al. (2015) to estimate the phase function of a bicontinuous medium is morealigned with the strategy used to estimate the extinction coefficient, and could probably at least becompared to the current approach. Approaches used in other disciplines (scattering in any porous medium) may also provide interesting alternatives (see for instance Haussener et al., 2012). In addition, it would be great to provide the values of the asymmetry parameter computed for the estimated phase functions,

which would allow comparison with usual assumptions made on this hardly measurable quantity.

**Response:** We agree that the grain segmentation process and selection of individual grains to be included in the phase function sampling can be subjective. Upon further consideration and reflection and in response to comments from the other reviewer, we have decided to take your advice and use an approach more similar to Xiong et al. 2015 and Haussener et al. 2012. This approach better utilizes the intact snow sample representation allowed by X-ray microtomography. In this approach, which we refer to as the localized phase function, the phase function is computed through ray-tracing by binning the reflected and scattering angles at each air/ice boundary, and then integrating the binned power to get the phase function (I.e., equation 12 in this paper).  Note that this method assumes a perfectly flat surface at the ice-air or air-ice boundary and ignores the aggregated effects of internal reflections within a particle on directional scatter. Since we are no longer computing the phase function for individual grains (i.e. the whole particle phase function method), we have cut figures 1 & 2 from the paper, and the text surrounding grain segmentation.  Overall, this change substantially improves our results when compared to observations. We believe this is due to an increase in forward scattering over the previous computation method. We also note that the newly computed phase functions compare well to the "P11" functions shown in Xiong et al. 2015 and Haussener et al. 2012.

This localized phase function approach also helps to address comment #3 regarding the use of $F_{ice}$.  In considering both the phase function and the extinction coefficient as determined by intersections with particle facets, rather than whole particles, we believe our approach of continuously depleting energy across the fixed distance is consistent with the larger model framework as the distance traveled each model integration is consistent with the distance between air/ice interfaces, rather than the distance between individual particles. This is addressed in more detail in the response to comment #3.

In response to your final comment here, our asymmetry parameters for rendered grains were lower than is often reported in the literature (generally between 0.6-0.7), however our values for the idealized rendered spheres and hexagonal plates were consistent with previous literature (e.g., 0.89 for a sphere with a size parameter of 1000).

2) A method is proposed to estimate the extinction coefficient of snow from the 3D images, which is based on sampling the probability to be scattered or absorbed when traveling a certain distance from a variety of locations chosen randomly. This strategy seems acceptable, but does not reproduce the dependence of $\gamma_{ext}$ on SSA and density $\rho$ expected for a collection of individual convex particles, that is $\gamma_{ext} = \rho \cdot SSA / 2$. This simple relation has been used in recent, similar, studies, though

(Malinka, 2014, Picard et al., 2016). Although this deviation from the expected behavior is already seen in the original paper by Xiong et al. (2015) (their figure 4c), it should be further investigated. For instance it would be worth checking that the present strategy reproduces the expected dependence in the case of a collection of spheres. If it were carefully confirmed that in real snow the relation $\gamma_{ext}$ = ρ · SSA / 2 does not hold, this would be an important result. Not also that Figure 14 should be recomputed, looking for a correlation between $\gamma_{ext}$ and the product ρ · SSA.

Response: This is an excellent comment and brings up an interesting issue which we had not considered. First, we have made a couple of minor adjustments to the curve fitting code to attain better curve fits for $\gamma_{ext}$, so the values are a little different, but still not consistent with ρ*SSA/4 (no diffraction). We had chosen to estimate the extinction coefficient following Xiong et al. 2015 to simply extend their approach from a 2D medium to a 3D rendered sample. In this approach, photons are initialized at random throughout the sample, both within and outside of the ice. Xiong et al. noted, without any further explanation, that this approach allowed for the inclusion of the microstructural features of the medium in the computation of $\gamma_{ext}$. We had not considered that this approach might not produce the $\gamma_{ext}$ = ρ*SSA/4 relationship, which implies that the incident radiation is initialized outside of a particle (i.e., the geometric scattering cross section is simply the particle projected area). Accordingly, Xiong et al. does not reproduce the $\gamma_{ext}$ = ρ*SSA/4 either (they also ignore diffraction).

To test this, we computed $\gamma_{ext}$ following the Xiong et al. method applied to a collection of spheres with specified SSA and density, and found that these computed values match closely to the values of Xiong et al. (e.g., 4.58 mm$^{-1}$ vs. 4.66 mm$^{-1}$ for SSA = 32.7 and ρ = 250 and 960 mm$^{-1}$ vs. 940 mm$^{-1}$ for SSA = 6.54 and ρ = 250), suggesting that this specific method for computing the extinction coefficient is more responsible for the deviation from the expected relationship, rather than particle shape.

We further tested this by computing solid and air chord lengths for the spherical particle meshes via random ray sampling and estimated $\gamma_{ext}$ following Malinka et al. 2014, since this expression for $\gamma_{ext}$ is in part derived from the $\gamma_{ext}$ = ρ*SSA/4 relationship. This resulted in $\gamma_{ext}$ much closer to the ρ*SSA/4 relationship (e.g., 0.48 mm$^{-1}$ for SSA = 6.5 and ρ = 250, as compared to 0.96 mm$^{-1}$ following Xiong et al.). This discrepancy (i.e., tendency towards higher extinction coefficients) may help address your comment on $F_{ice}$ and continuous absorption, since the extinction coefficient tends to higher values using the Xiong et al. method due to the particle initialization scheme.

We are aiming to add the following discussion on this point to a revised manuscript:

"Foundational work on light scattering in a collection of weakly absorbing particles indicates that, ignoring diffraction, $\gamma_{ext}$ is given as $\gamma_{ext}$ = ρ*SSA/4 (e.g., Van de Hulst, 1957, Kohkanovsky, 2004). This is substantially different than the relationship found between $\gamma_{ext}$, SSA and ρ illustrated in figure 14. We speculate that this is because the method for

*determining γext described in Xiong et al. (2015) and extended into three-dimensions here,*
*initializes particles randomly throughout the sample which relaxes the assumptions*
*regarding particle projected area implicit within the $\gamma_{ext} = \rho*SSA/4$ relationship.*"

Finally, we have taken your suggestion and computed the correlation between $\gamma_{ext}$ and the product of $\rho_{snow}$ and SSA for Figure 14, and have obtained substantially cleaner results. Thank you for an excellent suggestion. The written and visual results are now much cleaner and understandable.

3) To my knowledge, the use of the mean fractional ice path $F_{ice}$ is quite unique to this study. Generally, the medium is instead represented by its absorption efficiency, or equivalently by its single-scattering albedo. In Eq.18, the energy of the photon packet is decreased along the path $s$ due to absorption within the ice. However the distance traveled in the snow is underestimated when using a continuous representation, because the distance $l_{ext}$ between scattering (or absorption) events is less than the actual distance traveled, which should include internal reflections not accounted for in $l_{ext}$ (see Malinka 2014). As a consequence I believe (this has to be verified) absorption is overall underestimated with the present method. This can be easily corrected by carefully taking into account internal reflections (as is made for the estimation of $F_{ice}$). Hence I'd encourage the authors to compare their approach to a more standard one based on single-scattering albedo (in which case absorption is seen as a probabilistic localized event rather than a continuous process). This would help validate the $F_{ice}$ approach or point to fundamental and critical differences between both approaches. Also, it should be physically explained why a linear relation is expected between ρ and $F_{ice}$.

**Response:** This is another interesting comment, and while we cannot be certain that we are under-representing the distance traveled within ice, we think that $F_{ice}$ is correct in this specific context. We believe that the justification for the use of $F_{ice}$ is partially provided in the responses to comments #1 and #2.   From comment #1, when using a localized scattering perspective (as is done in Xiong et al. / Haussener et al.), the scattering direction is determined by scattering at an air/ice boundary, rather than a particle, which characteristically increases scattering in the forward direction over that of scattering computed for a specific particle shape.  Additionally, the Xiong et al. method for computing the extinction coefficient generates extinction coefficients that are broadly greater than those estimated by $\rho*SSA/4$.  It is tricky to understand exactly why this is the case, but we suspect that it is due to the snow "particle" perspective from which $\rho*SSA/4$ is derived as compared to the semi-random two-phase medium used by Xiong et al.  We speculate that combined with these approaches, the use of $F_{ice}$ is appropriate here.

To support this assertion, we performed a sensitivity simulation for a collection of spheres with d=0.6 whereby we used an estimated single scattering albedo through ray-tracing following Grundy et al.: Single_Scat = 1 – absorbed/total.  We combined this with an analytic extinction coefficient of $\rho*SSA/4$, and the phase function of a sphere at 1000nm.  We used

these optical properties to run the medium model to get spectral albedo, and compared that to the method that uses the extinction coefficient from Xiong et al. and F$_{ice}$ with continuous absorption (see results in figure below).  Note that in this method, the phase function is localized as in Haussener et al. 2012 / Xiong et al. 2015.  While not exact, these methods produce results that are generally close to eqs. 9/49 from Kokhanovsky and Zege 2004.  We suspect that localizing the phase function helped make the methods for generating the optical properties consistent with each other, and the medium model, and that our earlier more substantive differences can be traced to this.

Finally, we note that a linear relationship between ρ and F$_{ice}$ is a good approximation, at least for densities greater than 150 kg m$^3$.  The simplest physical explanation for this is that as density increases, there is simply more ice available for the photon to travel through. Assuming, that the proportion of internal reflections taking place throughout the sample is similar across the samples, then one would expect a generally linear relationship with density.  This fits with the equation you've provided that relates F$_{ice}$, density, and B.

[Figure]

4) Related to the previous point, I stress that some work has been focused in the past decade on the estimation of the absorption enhancement parameter B of snow (Libois et al., 2013, 2014, 2019), which is actually directly related to the proposed definition of F$_{ice}$. Estimations of B from the joint values of F$_{ice}$ and ρ provide results quite different from those obtained elsewhere, which at least

deserves comments, if it does not help pointing to deficiencies of the present study. Although this could in itself form relevant material for an independent, complementary study, it appears necessary to at least clarify this point.

**Response:** This is a great comment and we have additionally responded to a similar technical comment below. We hypothesize that our results aren't actually significantly different than the results elsewhere, but rather the apparent discrepancy is associated with the non-linear relationship between $F_{ice}$ and B and photons that do not accurately sample the snow sample during the ray-tracing algorithm. For instance, a track that interacts with only a single particle and then exits the sample after some internal reflections, $F_{ice}$ approaches 1, as compared to a track that traverses many grains yielding an $F_{ice}$ value that is more consistent with the sample density. Further, because B is a non-linear function of $F_{ice}$, any shift towards higher values of $F_{ice}$ due to these odd tracks would lead to a much larger than expected B parameter. If, we instead compute B from Equation 4 in Libois et al. 2019, instead of from $F_{ice}$ and density, we obtain values generally between 1.2 and 1.6. We agree that a deeper dive into looking at how B varies with $F_{ice}$ for different snow samples and types would make for an interesting complementary study.

5) The paper tries to differentiate from previous similar studies by focusing on transmissivity, in complement to more widely explored reflectance properties. However, only a few observations (practically, 3 reflectance spectra) are used to validate the model, which do not correspond to the primary focus of the work. Since these measurements are not very well simulated (compared for instance to those reported by Carmagnola et al., 2013), it does not give confidence in the transmissivity simulations, in particular in the near-infrared. We also note than in the visible some differences remain that could be partly explained by the presence of light-absorbing impurities, which is not really discussed.

**Response:** We have spent considerable time and effort in reframing the paper to include more observational comparisons. We had initially chosen to limit this evaluation to keep the paper length manageable and focused on introducing the framework. However, reworking the model phase function (see comments #1 & #3) and eliminating the grain segmentation, has substantially shortened the methods section which allows for a greater focus on observations. Further, the results obtained with a localized phase function computation more like the Xiong et al. method produces substantially more accurate results in the NIR. Finally, we have recast some of the wording to indicate less of a focus on transmissivity than the initial manuscript, while retaining an analysis and discussion on transmissivity.
   Lastly, the snow was clean and free of contamination. Specifically, we can say with some confidence that the snow was not contaminated with dust or black carbon, which is clear in the measured visible spectrum.

6) One of the main problems of this paper is that it lacks a proper, critical discussion. In the current version the discussion is 15 lines long and does not really question the whole results of the paper. I'm convinced that given the uncertainties arising from the chosen method to estimate local optical

properties, the issues related to the spatial representativity of the very small samples, and the very limited number of observations that unsuccessfully try to support the model, a much longer discussion would be very useful.

**Response:** In light of many of the larger issues that you brought up in your thorough review, we have made large changes to the paper, including in the scope, methods, and focus. Accordingly, we have included a more robust discussion to better contextualize our results and uncertainties with some of the suggested literature. Including discussion on the relationship between $F_{ice}$ and the B parameter, the relationship between $\rho*SSA$ and $\gamma_{ext}$ as compared to the relationships discussed in Kokhanovsky and Zege 2004, and sample representativeness of the snow layers.

**Technical corrections**

l.3 : "based on X-ray microtomography" is unclear → "reflectivity of snow samples based on X-ray microtomography images ?

**Response:** Updated to "*reflectance of snow based on X-ray microtomography images,..*" as suggested.

l.6 : is really the focus more on transmissivity than albedo?

Response: See response to Specific Comment #5 above for further details. We have reworded this sentence to "... *this study's effort is focused on simulating reflectance and transmittance in the visible and near-infrared (NIR) through thin snowpacks...*".

l.7 : sub-nivean hazard detection is mentioned in the abstract but not later on

Response: We thank the reviewer for pointing this out. As this is not a focus for this particular paper, we have removed it from the abstract. However, we briefly link this application to the 5% transmissivity contours shown in Fig 12 in the discussion.

l.8 : should snow grain size be replaced by SSA? Because the advantage of having 3D images of snow samples is to get rid of the simple, unrealistic, granular approach

Response: We have replaced grain size with SSA.

l.10 : not clear whether in the field transmissivity and/or reflectivity measurements were performed

Response: We only refer to the reflectance data in the results and so have updated the text here to specify that.

l.13 : "is limited to the top 5 cm" and "can penetrate" is awkward. Transmissivity is a property. Should be rephrased.
l.14

Response: We agree and have rephrased to make the point more clear. "...*indicate that snow transmittance is greatest in the visible wavelengths, limiting light penetration to the top 5 cm of the snowpack for fine grain snow but increasing to 8 cm for coarse grain snow.*"

l.15 – 15 : I think this result is quite obvious. Maybe consider providing another more specific result.

Response: We agree and have replaced this sentence with the following: "*These results suggest that the 5% transmission depth in snow can vary by over 6 cm according to the snow type*"

l.28 : "aggregate" is unclear

Response: While we were referring to spatially-averaged, we believe that is implied and have simply removed the term.

l.42 : to state that geometric optics works well some hint should be given about the typical size of scatterers in snow (or referring to a paper stating why it works well)

**Response:** We have modified this sentence to read:

"*The scattering of visible and NIR light in snow is well approximated with the geometric optics approximation because snow particles are much larger than wavelength (e.g., Kokhanovsky and Zege, 2004).*"

l.43 : I think the interaction between a "snow particle" and light is far from being well understood. Because we essentially don't know what a snow particle is. Consider rephrasing the sentence.

**Response:** We have rephrased this to: "*While the physics behind scattering and absorption are well understood for the geometric optics limit, …*"

l.47 : Here Mie theory is mentioned which seems to contradict the fact that geometric optics is used. Outside of the Mie regime, using a Mie code is probably useless.

**Response:** We have clarified this sentence to state that, as a simplification, snow is often represented as a collection of spherical particles and that Mie theory is used to determine the albedo in this case.

l.49 : I'd say that Mie theory CAN ONLY BE (by definition) applied to spherical particles

**Response:** We agree and have reworded these sentences to make that assertion clearer.

l.53 : please double-check that the errors highlighted by Dang et al. (2019) indeed result from the spherical hypothesis, and not from the two-stream approximation

**Response:** Thank you for calling our attention to this. Since we cannot confidently parse out the source of error in Dang et al. (2019), we have removed this sentence and instead focus on the variation in single scattering properties with grain shape as noted by Ishimoto et al. (2018) and reference additional studies attributing errors to the spherical assumption (Picard et al. (2009), Libois et al. (2013), Neshyba et al. (2003)).

l.59 : Is it similar to the approach of Malinka (2014), which could then be cited as well?

**Response**: We did not cite Malinka (2014) here as our focus was on previous ray tracing studies and Malinka (2014) uses a stereological approach. We will specify this at the beginning of that paragraph and plan to mention Malinka and several other studies in additional text in the introduction.

l.62 : I'm not sure in Dumont et al. (2021) they used ray-tracing on numerical samples to simulate optical properties.

**Response**: The editor has requested that we include this reference here.

l.65 : I'd encourage the authors to use reflectance/transmittance (the measured quantity) or transmissivity/reflectivity (the material property), but not a mixture of both.

Response: Thank you for pointing out the inconsistency. We have adjusted our text to use reflectance/transmittance.

l.72 : I understand that the main difference of this study with the previous ones upon which it builds is the special focus on transmissivity. If it's the case, consider being even more specific on this point.

Response: See response to Specific Comment #5 above for further details.

l.75 : what does "semi-quantized" mean?

Response: The phrase "semi-quantized" is in reference to the fact that the model tracks individual photon packets in a quantized method, however it uses a continuous depletion of energy through $F_{ice}$: Absorbed = $1 - exp (-\kappa_\lambda s F_{ice})$.

l.89 : "RT through non-spherical properties" is unclear. Do you mean to simulate single scattering properties of such particles?

Response: Yes, we are referring to single scattering properties of non-spherical particles. We have updated the text to clarify.

l.83 : again, is the reference to Dumont et al. (2021) appropriate?

Response: Again, the editor has requested that we include this reference here.

l.90 : remove "medium"

Response: Removed, thank you.

l.92 : how does this resolution compare with previous studies? Is it estimated to be sufficient to represent small-scale snow features that can have an impact on snow optical properties?

**Response:** Other snow optical studies that have employed micro-CT data (Donahue et al., 2021; Ishimoto et al., 2018; Gergley et al., 2013; Kaempfer et al.2007 ) have used voxel resolutions ranging from 7 to 175 microns, depending on the minimum grain size being imaged in the study, 0.025 mm to 0.5 mm in the case of previous studies. The grain size of the smallest samples we collected was estimated to be between 0.25 and 0.5 mm as determined from measurements of grain sizes made in the field using a hand lens and standard grain size measurements. Based on these estimates of grain size made in the field with the hand lens, the resolution of the micro-CT, at 20 microns, is roughly on the order of 10 times (12.5 times) the linear size of the minimum grain size we are imaging. We used the commonly employed Nyquist sampling criterion, which requires a minimum of 2.3 pixels per linear feature, to determine that the resolution was sufficient for the grain sizes we had sampled. The Donahue et al. 2021 study used the same resolution for grain sizes similar to our study (the authors also used the same make and model micro-CT, a Skyscan 1173, that we use).

l.92 : does "1D" mean plane-parallel, that is horizontally homogeneous layers? Maybe clarify this

**Response**: Yes, we have clarified.

l.99 : Doesn't the 1D model need single-scattering albedo $\omega$? Would it be more appropriate to introduce first the 1D model (reverse 2.1 and 2.2) to highlight what properties are needed? Maybe consider to make the link between $F_{ice}$ and $\omega$ as follows (valid for weakly absorbing media):

$F_{ice} = B \cdot \rho/(\rho_{ice}+(B-1)\rho)$, where B is the absorption enhancement parameter (see Libois et al., 2019) And $1-\omega = 2B\kappa/(\rho_{ice} SSA)$ (Picard et al., 2016)

**Response**: We do not think that the medium model needs single scattering albedo since energy is depleted using $F_{ice}$, and not a medium absorption coefficient. There is some nuance in computing the B parameter directly from the mean $F_{ice}$ and the sample density (see responses to other comments) that makes it difficult to estimate single scattering albedo directly from $F_{ice}$ as described above. However, we can estimate the single scattering albedo directly from the ray-tracing following Grundy et al. 2000. That said, because the Xiong et al. method for estimating the extinction coefficient yields substantially different $\gamma_{ext}$ than the $\rho*SSA/4$ (see responses to other comments), the single scattering albedo estimated from following the above equation would be inconsistent with this framework.

l.102 : unclear. Should it be the distance between extinction events (which can either be scattering

or absorption)? And the relation is simply the inverse, no? Could you use there the more straighforward expression of extinction coefficient as density x SSA / 2 (depending on whether you include diffraction or not in scattering). See for instance Malinka (2014).

Response: We have reworded this sentence to be the following:

"*In considering a photon of light traveling through the snow medium along a path, the photon is considered extinct when it intersects and is scattered along an air/ice boundary or is absorbed within the ice. The extinction coefficient is then inversely proportional to mean distance traveled between these scattering and absorption events.*"

l.115 : could you comment on this wavelength dependence. Is it a default of the method, or something expected?

Response: It's a default of the method (see Xiong, et al. 2015), and we expect it since the extinction coefficient includes both scattering and absorption events. However, in practice, there is no wavelength dependence until ~1300nm when the ice absorption coefficient is larger. This fits with the weakly absorbing particle assumption that yields $\gamma_{ext}=\rho_s*SSA/2$ (i.e, no wavelength dependence).

Eq. (4) : typo sign error; also, sin and cos should not be italic (same think throughout the paper)

Response: We corrected all, thank you.

l.137 : add $v_r$ and $v_t$ in parentheses here, and remove the end of the sentence l.141

Response: Thank you for pointing this out, we have edited as suggested.

l. 142 : this sounds very awkward to mention individual snow grains here, while the advantage of working with X-ray microtomogography images is to get rid of the particular representation of snow. Also, it is obvious that isolating grains from such an image is very arbitrary and isolated grains can behave very differently than the same grains being slightly sintered. Definitely it's not tricky to define the phase function of a porous medium based on local characterization (see for instance Haussener et al., 2012). This deserves more caution. The approach of Xiong et al. (2015) might be more appropriate than isolating "snow grains". An illustration of the segmentation process would be useful if this strategy is maintained. Could you also provide (where it best suits) the values of the asymmetry parameters obtained with this approach?

Response: Upon further reflection, we have decided to take your suggestion and use a local phase function characterization following closely to Xiong et al. and Haussener et al., 2012. This approach eliminates the subjective method of isolating grains, improves our results, and fits better with the broader plane-parallel framework that uses $F_{ice}$ instead of an absorption coefficient. See responses to comments 1 and 3 for more elaboration and see below for an updated figure comparing the spectral albedo for the fresh snow.

[Figure]

l.170 : this grain selection seems very arbitrary and would deserve more attention or explanations. Also, how many grains should be averaged to have something representative? What's the variability of the phase function across grains from a same sample?

Response: We agree that the grain selection is somewhat arbitrary. While no longer wholly relevant since we have removed this section from the methods (see responses to other comments), we had been choosing grains to ensure that they were not located on a sample boundary and had no clearly artificial boundaries. The number of grains averaged to generate the phase function were dependent on the sample. For example, some of our very large grain samples only had a handful of grains to average, whereas the fine grain samples had several hundred grains to sample from.

l.172 : some comment is needed on the relevance of using the properties of a very small sample to represent a whole (necessarily heterogeneous) snowpack. Said differently, what is the representativity of the sample?

Response: We agree on this point and have added some quantitative content in the discussion section to address this point.

l.175 : reference to Picard et al. (2016) might be relevant

Response: We have added a reference to Picard et al. (2016).

l.176 : problem with the beginning of the sentence

Response: Fixed, thank you.

l.198 : could you double-check the equation for $\mu_y$. There might be a sign error. Also didn't you forget the last terms for $\mu_x$ and $\mu_y$?

**Response:** In revisiting this formula as it relates the cartesian directions to the scattering angle, we do not see a sign error for $\mu_y$. However, the last terms for $\mu_x$ and $\mu_y$ were left out of the text and have been added. Fortunately, this oversight was limited to the paper text, and the code in the model was correct. Thank you for catching this!

l.202 : is this approach the initiative of the authors, or was it taken from another paper? I'm afraid it's wrong because it overall underestimates the total distance traveled by a photon (the distance

traveled between scattering events does not include enhancement in the ice phase), hence the total absorption. This is probably tricky.

**Response:** This approach to use $F_{ice}$ to continuously deplete energy was the initiative of the authors (as far as we know). We agree that this is tricky, however, we think that the use of $F_{ice}$ instead of a medium absorption coefficient is consistent with the framework whereby the extinction coefficient is computed using the probabilistic form of Beer's Law at varying distances from the particle initiation. We note that using this method, the scattering coefficient is larger than expected by $\rho*SSA/4$ for rendered spheres by a factor on the order of 1.3 - 1.8. We expect that this scattering coefficient is consistent with the $F_{ice}$ based continuous energy depletion. Finally, the lack of absorption noted in the original paper was entirely corrected once localizing the phase function such that it was oriented at air/ice boundaries, rather than to individual segmented grains. We did a brief comparison of methods to support this hypothesis. Please refer to the more detailed response for specific comment #3 provided above.

l.207 : the "Russian roulette" should probably be better explained. What happens to the photon packets that are not killed?

**Response:** Photons that are not killed are provided additional energy proportional to "*m*" which supplies a small bit of additional energy to the model to account for the energy removed when a photon is killed. We added some additional text that we hope clarifies this point.

*"In this formulation, energy removed from the model by killing photons is compensated for by adding a comparable amount of energy to the photons below the prescribed threshold which are not killed, thereby preserving statistical energy conservation."*

Figure 3 : labels should be larger (also for Figs. 9, 10, 13). Why using markers? Why not simply having two layer with different shades?

**Response:** We will increase all axes labels and tick marks for the noted figures. For Figure 3, we felt that the markers better allow the reader to imagine the photon hitting and interacting with individual air/snow interfaces, as occurs in a real snowpack.

l.227 : I did not understand "is used to simplify the complex reflectance properties of a rough surface"

**Response:** We have simply removed this text as it did not add a whole lot to the sentence. Thank you.

l.238 : not clear whether samples are taken from the surface or in the pit (to sample various layers)

**Response:** The samples were taken from the surface, 0-7 cm, in the case of the small grained

sample, and from 14-19 cm depth in the case of the large grained sample.

l.241 : how later (compared to snow sampling) were the images taken?

Response: The small grained sample was imaged 18 days after snow sampling, while the large grained sample was imaged 53 days after snow sampling due to limits in access to the laboratory due to COVID-19 concerns. All samples were stored at -30 degrees C to limit metamorphic change in the intervening timeframe.

l.245 : "optimization" is unclear

Response: The FieldSpec 4 requires optimization which adjusts and improves the detector sensitivities for the probe and light source currently in use. We have included this clarification in the text.

l.247 : was it nadir observations?

Response: Yes.

l.255 : what's the size of the aluminium panel? How was it practically inserted? How did you ensure that it is horizontal under the snow? What's the precision of the depth position (transmissivity greatly varies with depth)? 4.75 cm seems very (too) accurate for a depth measurement! Typo for "aluminium"

Response: The aluminum panels were 16x16 inches. They were inserted into a snow pit sidewall, carefully to disturb the above snow as little as possible.  In practice we started from the bottom and worked up through the snowpack so the "slot" left behind after pulling out the panel wasn't included in the following measurement.  We had no way to ensure that it was perfectly horizontal beneath the snow, though we think a likely bigger contribution to variations in depth between the panel and the snow surface are minor (<0.25cm) variations in the actual snow depth.  Snow depth measurement precision was likely on the order of 5mm, but considering minor fluctuations in snow surface depth, or the possibility of subtle tilts in the panel, we have revised this precision to 10mm.  With regards to the typo, "aluminum" is the typical US spelling and we will leave it to the editor to select which spelling is preferable within this journal's standard.

Figure 4 : maybe not very useful.

Response: We have cut figure 4 from the paper.

Figure 5 : An indicative scale would be helpful

Response: We have included the size of the panels in the text. However, we disagree that adding a scale at this point to the image will be beneficial since it would require a

considerable approximation, would likely be visually unappealing, and wouldn't provide any additional information.

l.271 : Is the cylinder still 7 cm high for the detailed analysis, or limited to a cubic sample?

Response: As noted on l.285, a cubic sub-sample is used for the detailed analysis.

l.281 : totally agree, and this should be further discussed. Does this literature correspond to optical studies? If not, I see no reason that this approach is satisfying for optical issues.

Response: We agree that there is some amount of subjectivity in the grain segmentation process. In fact, improving grain automated segmentation from uCT for both optical and structural mechanics is an ongoing topic of research we are pursuing. We also note that Ishimoto et al. (2018) used a subjective grain segmentation algorithm on snow as part of an optical study, although they didn't use watershed. However, in response to previous comments, we have recast the phase function to be localized (e.g., Xiong et al., Haussner et al.) and no longer rely on segmented grains. So, this subjectivity is no longer relevant to this specific study.

l.313 : what is the underlying albedo? Visible light can indeed reach the ground

Response: Underlying surface albedo was treated as a 100% absorptive lower boundary. Transmissivity was simulated at ~ 1% for the visible wavelengths.

l.317 : "of the each"

Response: Typo has been fixed, thank you.

l.318 : how is albedo computed? What are the illuminating and viewing angles?

Response: The illumination was downward diffuse (e.g., random incident vector directions). Albedo is approximated by simply tracking the ratio between the photon energy escaping the top of the model over the input energy. While it is more correct to compute it by hemispherically integrating the BDRF / DCRF, it's quite a bit more computationally expensive to do so, and we find the differences between these two methods were negligible until higher zenith angles.

l.320 : "that favors" reads awkward → "it shows the strong sensitivity of NIR albedo to snow microstructure"

Response: We have reworded as suggested.

l.323 : it's more a dependence than a relationship

Response: We have further specified the relationship as a dependence, as recommended.

l.325 : is exponential qualitative or confirmed? Consider referring to Eq. 9 of Kokhanovsky and

Zege (2004)

**Response:** This was qualitatively similar to what is seen in the literature. We have added citations to that effect (e.g., Xiong et al.). Further, we show the plot from eq. 9 from Kokhanovsky and Zege to contextualize the results noting that eq.9 approaches an albedo of one for all wavelengths more rapidly due to the cos(zenith) -> 0, and that our results are more consistent with those from Xiong et al., particularly for the NIR wavelengths.

Figure 10 : 25000 photons per wavelength (how many by the way?) or for all wavelengths?

**Response:** 25000 photons per wavelength.

l.326 : what "observed" behavior? Your work or from the literature?

**Response:** We have clarified this sentence to read:

*"consistent with the results from previous studies on the relationship between snow albedo and zenith angle (e.g., Li and Zhou, 2003; Kokhanovsky and Zege, 2004;  Xiong e al. 2015).*

l.326 : "with good fidelity" is not justified. What do you compare to? More generally your model could not but reproduce that, so it's not a proof of the model being "good".

**Response:** Agreed, we have reworded this sentence. "*This indicates that the model is capable of reproducing the surface anisotropy of snow described in the literature (e.,g Aoki et al. 2000; Hudson et al. 2006; Kaempfer et al. 2007; Dumont et al. 2010; Xiong et al. 2015).*"

Table 1 : would be useful to provide the parameter B as well (see Libois et al., 2014). In particular the values would be 1.95 for fine grain and 2.05 for coarse grain (TBC). Also $\gamma_{ext}$ is quite different from $\rho \cdot SSA/2$ (see Libois et al. 2013 or Picard et al., 2016), which would certainly deserve some comment. Have you applied your technique to a collection of isolated spheres (with known $\rho$ and SSA) to check whether you can obtain the exact (known) value of $\gamma_{ext}$?

**Response:** To answer the first part of this question, computing B from mean $F_{ice}$ and sample density here is a little tricky, and we suspect isn't providing an accurate answer. We think that this is because B increases non-linearly as a function of $F_{ice}$ for a given density and, as a result, the mean $F_{ice}$ (or mean B parameter) for several different photon tracks through the medium would be weighted towards a higher-than-expected value. Specifically, individual B values for photon paths with $F_{ice}$ less than the average would be outweighed by overly high B values for higher $F_{ice}$ values, which we expect are photon paths that enter and exit the medium rather quickly interacting with perhaps only 1 "particle." We can show this by comparing sampled histograms of $F_{ice}$ to histograms of B, and indeed, the mean value of B

is shifted to the left of the most frequent value, which is typically in the 1.3-1.6 range (see figure below). We propose a better way to estimate B from this framework is to follow equation 4 from Libois et al. 2019, and simply compare the solid ice fraction from a photon path that reflects and refracts throughout the medium to the solid ice fraction following a straight chord through the medium. Following this method, we get values closer to those reported within the literature, and we find that that B converges to 1.26 for a collection of spheres. We suspect that these two methods do not yield equivalent values for B because the sample density may not be representative of the density for each photon track that passes throughout the medium. Please see our response to an additional comment below for more information on this point.

[Figure]

Red line = B from mean Fice, Black dashed line = B = 1.26

Yes, we have computed extinction coefficients for several collections of spheres. Please see response to specific comment #2 above for details.

l.332 : it's not clear what transmissivity is here. It it the flux at a particular depth within the snow layer of 20 cm? Or it it the transmittance of a snow layer of thickness X (with black surface beneath), where X is varying. Without precision I'd assume it's option 2. This makes a big difference and should be clarified.

**Response:** It is option 2, we have made some clarification in the text to clarify this:

*"Finally, we use the model to provide an initial assessment of the impacts of snow microstructure on simulated spectral transmittance at specified depths within a homogenous snowpack."*

l.336 : an optical thickness is unitless. The penetration depth is a length, though

**Response:** Fixed, thank you for this correction.

Figure 12b: shouldn't the lines be removed, and the red line added to 12a? Because they correspond

to transmissivities, not differences in transmissivities

**Response:** We have chosen to keep the lines on 12a, since they correspond to the penetration depth for each simulation, and we think that the difference in penetration depths nicely compliments the difference in transmittance show in the shaded contours.

Table 2 : as for Table 1, I found B values ranging from 2.3 to 2.8, which are somehow much larger than previous estimates. Again $\gamma_{ext}$ is quite different (30%) from $\rho \cdot SSA/2$ (or even $\rho \cdot SSA/4$ if diffraction is not considered). Also, depth seems to start from the ground, which is surprising (looks more like height). Finally, I doubt that SSA estimation can reach 0.01 m² kg⁻¹ precision.

**Response:** We have updated the SSA values in the table to be precise down to the 10, not the 100s decimal place. Please see above comment for table one discussing the $\rho*SSA/4$ relationship and differences between how the scattering (extinction) coefficient is computed here vs. elsewhere. Again, see additional comments related to computing B from mean $F_{ice}$ vs. computing it directly from the ray-tracing following eq. 4 from Libois et al. 2019.

l.345 : why not using the 4% reflectance mentioned earlier?

**Response:** This may be a mistake in the paper. Regardless, we have rerun the simulations with the 4% reflectance lower boundary. Thank you for catching this!

l.347 : I don't understand what "remarkably good" means, given that substantial differences are seen in Fig. 13. Why are the differences in the NIR so large?

**Response:** We have reworded this to be quantitative. Specifically, we have quantified "good" by stating the RMSE for the visible (300-850) and NIR (850+) ranges, in addition to the figure. We have also updated this result to reflect the change in the phase function, which yields a much better result, and especially improved the comparison in the NIR. Thank you for the suggestion.

Figure 13 : why is there a line in between points?

**Response:** We feel that the lines help make the figure easier to read and more visually appealing, and have chosen to keep them.

l.350 : errors in the phase function are mostly impacting the NIR, but they also affect the visible range, which is why it deserves much more attention.

**Response:** We agree, and we have replaced the particle phase function in the framework with a localized phase function that does not rely on grain segmentation or individual particles, more like the phase function in Xiong et al. & Haussener et al. This phase function approach is more consistent with the approaches for determining the extinction coefficient and the absorption within the snow. Accordingly, this change alone results in dramatic improvement in model performance.

l.363 : according to the formula presented above ($F_{ice}$ = B · ρ/(ρ$_{ice}$+(B-1)ρ)) it's no surprise to have a linear-like behavior in the explored range of snow densities. However the 0.25 residual is surprising. I'd be worth computing B for all available data based on your approach.

Response:  See above comment regarding how we chose to compute B.  In response to your suggestion, we are working on calculating B for all of the samples following eq.4 from Libois et al., 2019. We note that while the relationship between Fice and density is well approximated by a linear function for the mid-range densities, there is a visible convex inflection in the scatter plot, which is consistent with the relationship between Fice and density in the above formula.  The 0.25 relationship is interesting, and we suspect that this is due to a general propensity for the mean $F_{ice}$ averaged over all photon tracks to fall to the right of the most frequent value of $F_{ice}$ in the histogram (see figure below).  We suspect that this is due to some photon tracks that interact with only a single particle within the same, and bounce out, leading to $F_{ice}$ that approaches 1.  Accordingly, the linear relationship between $F_{ice}$ and density appears as expected, though is shifted towards higher values than would be expected, leading to a y-intercept of 0.25.  This is likely related to your other comment regarding the abnormally high B values.  Note that in the below histogram, if the peak of the histogram is used (0.35) instead of the mean (0.5), the B parameter for the sample density of (275) is ~1.26 vs. 2.4.

[Figure]

l.372 : this has been known for a while (Eq. 23 of Bohren and Barkstrom, 1974)

Response: This section will be updated and revised based on the major comments above. We will ensure that the relevant references are included in that discussion.

l.368 : again, use the product ρ·SSA instead of a bilinear regression. Again a residual in such regression should be commented.

Response: We have replaced the bilinear regression with a regression against ρ*SSA, which is a much cleaner result. Thank you.

l.372 : what does a r² of 0.25 mean in terms of correlation?

**Response**: The correlation coefficient in this case is 0.5, meaning there is a positive correlation but it is not significant.

l.380 : suggest → confirm

**Response:** Corrected, thank you.

l.387 : can you clarify what "we pair" means

**Response**: We have removed the first part of this sentence as it is unnecessary and may be confusing. To be more succinct, it now reads "*We note that high values of $\gamma_{ext}$ are more likely to coincide with high values of $F_{ice}$ due to the shared dependence of these variaables on snow density in most snowpacks*."

l.390 : not clear why the highly scattering nature of snow explains the different sensitivities to $F_{ice}$ and $\gamma_{ext}$. I'd use an analytical expression of the penetration depth (see for instance Libois et al., 2013) to show that the dependence on $\gamma_{ext}$ is linear, while the dependence on B is square root

(changes in $F_{ice}$ are proportional to changes in B if density is unchanged). The ~3 scaling of $\gamma_{ext}$ results in a ~3 scaling of penetration depth. The ~2.3 scaling in $F_{ice}$ results in a ~1.5 scaling, which is consistent with your simulations.

Response: Thank you for another insightful comment. The original sentence on the highly scattering nature of snow was included to reflect that the most influential variable driving penetration depth within the snow was the one related to scattering ($\gamma_{ext}$) and not the one related to absorption ($F_{ice}$).  In our original reading of Libois et al. (2013), we had not made the analytical connection between the penetration depth and $\gamma_{ext}$.  In accordance with this comment, we have replaced this sentence to refer to the analytical expressions for the e-folding depth as a function of $\gamma_{ext}$ to show that a linear scaling is expected, and commented on the square root dependence of B:

"The simulated relationship between $\gamma_{ext}$ and penetration depth (L) is consistent with the analytical expressions showing a linear relationship between L and $\gamma_{ext}$ discussed in Libois et al. (2013).  Specifically, the factor of 3 decrease in the extinction coefficient corresponds to a factor of 3 increase in penetration depth.  Additionally, in showing a muted impact of $F_{ice}$ on L that increases with wavelength, the relationship between $F_{ice}$ and penetration depth is also consistent with the L ~ SQRT(B*kabs) relationship discussed in Libois et al. (2013). For instance, at 1300nm, the simulated increase penetration depth is proportional to square root of the increase in $F_{ice}$."

l.392 : never exceeds 2.5 cm, no?

Response: Yes, however this specific answer has changed once we localized the phase function.

l.393 : leading to **an** increase

Response: Great catch!  Thank you.

l.398 : for crust layers, how would you isolate grains from the 3D image?

Response: Using the same method as before, though with a larger minimum threshold distance for the Euclidian difference partitioning.  Certainty, this was a lot trickier to get realistic grains, and resulted in less grains available for the phase function computation. Which goes to one of your points made earlier.  However, because we have made the shift away from using isolated grains to compute the phase function, this is no longer relevant.

l.404 : at what wavelength were the phase function and $\gamma_{ext}$ estimated?

Response: 1000nm

l.407 : unit missing

**Response:** Fixed, thank you.

l.422 : I don't agree with "high accuracy"

**Response:** See response to specific comment #1. Improvements have been made to the model that result in better agreement between simulated and observed spectral albedo (Figure 13).

l.425 : at what wavelength?

**Response:** It is stated at the beginning of the sentence that we're referring to the visible portion of the spectrum here but we have further clarified by adding mainly in the 400 nm – 650 nm range.

l.426 : was the reduction of albedo mentioned earlier?

**Response:** We have rephrased to specifically reference transmittance, and not albedo, since an explicit relationship between albedo and transmission was not referenced earlier.

l.428 : again, this is obvious

**Response:** We have removed this sentence, thank you.

l.435 : consider reading Hagenmuller et al. (2019)

**Response:** Thank you for the suggestion.

*Additional suggestion* :

Here is a procedure to test the validity of the 1D ray-tracing code (although I don't particular believe it does not work):

- Consider an homogenous (horizontally semi-infinite) layer of thickness L, non absorbing
- Illuminate it with diffuse light
- Record the path lengths of escaping (reflected and transmitted) photons
- Check that the average path length equals 2L, **whatever the chosen phase function**
- If it does not work there is an issue somewhere
- See Blanco and Fournier (2003) for more details

**Response:** Thank you for this constructive suggestion to test the validity of the ray-tracing code.  In performing this test for a small variety of depths without absorption we find that the mean pathlength <L>/depth approaches ~2.  So, we believe the random walk ray-tracing code meets the criteria of this test.

**References**

Blanco, S., & Fournier, R. (2003). An invariance property of diffusive random walks. *EPL (Europhysics Letters)*, *61*(2), 168.

Bohren, C. F., & Barkstrom, B. R. (1974). Theory of the optical properties of snow. *Journal of Geophysical Research*, *79*(30), 4527-4535.

Carmagnola, C. M., Domine, F., Dumont, M., Wright, P., Strellis, B., Bergin, M., ... & Morin, S. (2013). Snow spectral albedo at Summit, Greenland: measurements and numerical simulations based on physical and chemical properties of the snowpack. *The Cryosphere*, *7*(4), 1139-1160.

Hagenmuller, P., Flin, F., Dumont, M., Tuzet, F., Peinke, I., Lapalus, P., ... & Charrier, P. (2019). Motion of dust particles in dry snow under temperature gradient metamorphism. *The Cryosphere*, *13*(9), 2345-2359.

Haussener, S., Gergely, M., Schneebeli, M., & Steinfeld, A. (2012). Determination of the macroscopic optical properties of snow based on exact morphology and direct pore-level heat transfer modeling. *Journal of Geophysical Research: Earth Surface*, *117*(F3).

Kokhanovsky, A. A., & Zege, E. P. (2004). Scattering optics of snow. *Applied Optics*, *43*(7), 1589-1602.

Libois, Q., Lévesque-Desrosiers, F., Lambert-Girard, S., Thibault, S., & Domine, F. (2019). Optical porosimetry of weakly absorbing porous materials. *Optics express*, *27*(16), 22983-22993.

Libois, Q., Picard, G., Dumont, M., Arnaud, L., Sergent, C., Pougatch, E., ... & Vial, D. (2014). Experimental determination of the absorption enhancement parameter of snow. *Journal of Glaciology*, *60*(222), 714-724.

Picard, G., Libois, Q., & Arnaud, L. (2016). Refinement of the ice absorption spectrum in the visible using radiance profile measurements in Antarctic snow. *The Cryosphere*, *10*(6), 2655-2672.

**Additional References**

Donahue, C., Skiles, S., and Hammonds, K. (2021), In situ effective snow grain size mapping using a compact hyperspectral imager. Journal of Glaciology, 67(261), 49-57., doi: 10.1017/jog.2020.68

H. Ishimoto, S. Adachi, S. Yamaguchi, T. Tanikawa, T. Aoki, K. Masuda, Snow particles extracted from X-ray computed microtomography imagery and their single-scattering properties, Journal of Quantitative Spectroscopy and Radiative Transfer, 209, 2018, 113-128, ISSN 0022-4073,
doi.org/10.1016/j.jqsrt.2018.01.021.

M. Gergely, F. Wolfsperger and M. Schneebeli, "Simulation and Validation of the InfraSnow: An Instrument to Measure Snow Optically Equivalent Grain Size," in *IEEE Transactions on Geoscience and Remote Sensing*, vol. 52, no. 7, pp. 4236-4247, July 2014, doi: 10.1109/TGRS.2013.2280502.

Kaempfer, T. U., Hopkins, M. A., and Perovich, D. K. (2007), A three-dimensional microstructure-based photon-tracking model of radiative transfer in snow, *J. Geophys. Res.*, 112, D24113, doi:10.1029/2006JD008239.

Li, S. and Zhou, X., 2003, July. Accuracy assessment of snow surface direct beam spectral albedo derived from reciprocity approach through radiative transfer simulation. In IGARSS 2003. 2003 IEEE International Geoscience and Remote Sensing Symposium. Proceedings (IEEE Cat. No. 03CH37477) (Vol. 2, pp. 839-841). IEEE.

Hudson, S.R., Warren, S.G., Brandt, R.E., Grenfell, T.C. and Six, D., 2006. Spectral bidirectional reflectance of Antarctic snow: Measurements and parameterization. Journal of Geophysical Research: Atmospheres, 111(D18).

---

## Author Comment (AC2)

The Cryosphere Discuss., referee comment RC2
https://doi.org/10.5194/tc-2021-310-RC2, 2021

[Figure]

**Comment on tc-2021-310**

Anonymous Referee #2

Referee comment on "A generalized photon-tracking approach to simulate spectral snow albedo and transmissivity using X-ray microtomography and geometric optics" by Theodore Letcher et al., The Cryosphere Discuss., https://doi.org/10.5194/tc-2021-310-RC2, 2021

This study develops and applies a Monte Carlo photon tracking model to simulate snow reflectance, using micro-CT scans of the snow as input. Because the micro-CT scans (necessarily) only apply to ~1 cubic centimeter domains, a hybrid approach is adopted to extend the optical properties of ice grains obtained from these samples to implicitly model the reflectance and transmittance of deeper, plane-parallel snowpacks. Comparisons of measured and modeled spectral albedo of snow with black targets placed at different depths was generally favorable in the visible portion of the spectrum, where the influence of the targets is strongest, indicating the simulations of bulk transmittance are likely accurate. Overall, this is a useful contribution to the literature, and the proposed approach has promise for broader application, but the issues described below should be addressed prior to publication.

General issues

Section 2, lines 90-97: The reasoning for applying these two distinct modeling approaches becomes clearer as one works through the manuscript, but I think the discussion here should be expanded to clarify the need and reasoning for these two separate approaches.

Related: One of the main attractions of explicitly applying a Monte Carlo model to 3-D snow images is that it potentially precludes the need to identify and distinguish distinct snow grains and treat them as independent scatterers. And yet, the approach applied here essentially does that, treating identified grains as independent scatterers, necessitated by the small sample size of micro-CT scans and excessive computational needs associated with tracking photons through a sufficiently large sample composed of 20um voxels. I do think the approach developed is innovative and useful, but I would like to see a clearer discussion of (a) potential biases and limitations associated with separating ice grains and treating them as independent scatterers, and (b) potential sensitivity of modeled results to the grain separation algorithm applied. I would think there is a fair amount of subjectivity in the identification of grain boundaries, especially in sintered snow, and I think it would be helpful to discuss the implications of this.

**Response:** In response to another reviewer who shared the same concern, we have decided to reframe the phase function such that grain segmentation and individual particles are no longer required, since the phase function was the only optical property that required grain segmentation. This method follows more closely to Xiong et al. 2015 and Haussener et al. 2012. Upon further reflection, we understand that this method is

more consistent with the estimation of the extinction coefficient and computation of absorption within the snowpack, and as a result substantially improves our results. However, this change does require major rewrites to the methods section, and changes in our results interpretation.

Line 116-119: What is the statistical uncertainty in the extinction coefficient derived with

this technique?

**Response:** The uncertainty for a given value of the extinction coefficient is very low for set parameters. For example, repeating the calculation of it for the same sample, same wavelength, and same curve fitting technique yields a standard deviation of <0.05 if enough photons are used (>2000 seems to be sufficient for convergence). However, there is more substantive uncertainty if the parameters that influence the curve fitting are modified. For example, if the distance sampling is different, or if the initial guess that feeds into the curve fit is modified can yield differences as high as 0.3.

Although the comparison between modeled and measured spectral albedo (Figure 13) is quite good in the visible, the discrepancies at wavelengths longer than 1000 nm are rather substantial. The authors speculate that this could be due to errors in the derived particle scattering phase functions, and indeed albedo in this part of the spectrum is strongly influenced by grain size and shape, so errors in the identification and rendering of individual grains could be responsible for this. Because the penetration depth of near-IR radiation in snow is very short, however, one alternative explanation is that grain morphology of the very top of the snow (e.g., top ~1mm) could be different from the mean morphology of the top 2cm, from which the sample was collected. Grenfell et al (1994) speculated that unresolved snow grain size of the top millimeter of snow could be responsible for similar discrepancies found in their study. Another potential consideration is uncertainty in the near-IR refractive indices of ice, as described by Carmagnola et al (2013) and Dumont et al. (2021). Even more serious than uncertainty in the near-IR refractive indices, however, is application of spectrally-constant refractive indices, as suggested on p.24. (Please see the next comment). Given the magnitude of the modeled albedo bias in the near-IR, I suggest expanding on the analysis and discussion of potentialunderlying causes of this.

**Response:** Thank you for the suggestions and relevant references. We have incorporated some discussion on these points to the manuscript. In particular, on the first point related to differences in the top 1mm of snow vs. the top 2cm. Specifically, we find that the physical and optical properties of a 1cc rendering of snow, are quite variable over short depths (e.g., 2cm of snow can have a range of optical properties depending on how the uCT renderings are subset). However, as noted above, our results are substantially improved by modifying the phase function such that it determines the scattering direction at individual dielectric boundaries, rather than "whole particle" scattering. We believe that this orientation is much more consistent with the overall framework, as it intrinsically assumes photons are traveling between scattering events, not particles.

Discussion on p.24 indicates that the authors assume ice optical properties independent of wavelength. I appreciate that this is done to reduce computational expense, but I believe this could lead to non-negligible biases, especially in the near-IR, and may even relate to the modeled albedo bias described above. The impact of spectral variations in ice optical properties can be seen in Mie solutions for ice spheres, which for 1000um spheres produce scattering asymmetry parameters ranging from 0.888 at 500nm to 0.915 at 1400nm, indicating differences in the scattering phase function that will lead to differences in modeled albedo. I think the importance of this issue should be probed more, potentially within the context of modeled albedo biases shown in Figure 13.

**Response:** As mentioned, this was done in accordance with computational expense, and this is often assumed that n_ice is constant throughout the visible and NIR at ~ 1.33 in other studies. As part of a more comprehensive discussion, we have added some comments on this, specifically related to the phase function, and include a new figure showing how the phase function varies according to wavelength.

Specific issues:

Line 38: "scattering of electromagnetic energy ... determined by the different refractive indices for ice and air" - Perhaps add "and geometry of the ice-air interfaces".

**Response:** Done, thank you for this suggestion.

Line 86: "Further, this framework ignores the wave properties of light, such as phase and diffraction" - I think it would be helpful to include a bit more discussion (i.e., 2-3 sentences) on just how important neglect of diffraction may or may not be. You could potentially draw on work from Liou et al (2011) and earlier work with co-author Yang, who include diffraction (at some level) in their derivation of optical properties for non-spherical ice particles.

Response: Thank you for this suggestion, accordingly we have added a couple of sentences further discussing this shortcoming:

*"While some work has been done incorporating diffraction into geometric optics scattering for non-spherical particles (e.g., Yang et al. 1997; Liou et al. 2011), because this framework treats snow as a two-phase medium rather than a collection of particles, accounting for diffraction isn't straightforward. Further, because the diffraction pattern is strongly forward scattering (Xiong et al. 2015), we anticipate that this simplification is appropriate here, though we acknowledge that diffraction may be more important for the longer NIR wavelengths."*

Line 99-100: Please include the units of these optical properties.

**Response:** Done, thank you.

Line 118: Just to clarify, the curve is fit to P_ext vs. L ? As commented above, what is the statistical uncertainty in the extinction coefficient derived with this technique?

**Response:** Yes, the curve-fit is from P_ext to L, through Beers law to get the extinction coefficient. The uncertainty is generally very small (<0.05), however by playing around with how the curve fitting is performed, it becomes a little larger, specifically by changing the sample distance (L) for calculating P_ext, and modifying how the curve fit is performed. See above response for more information.

Line 136: How is the vector normal of the ice surface calculated? Perhaps the reader could be referred to section 2.5 for more info, but immediate questions that come to mind are: Are grain boundaries represented as facets (like in Figure 1), resolved only to the voxel size, or as curved surfaces? If curved, is there a resolution to the derived curve, or is it a mathematical description that gives a precise surface normal for any point of ray intersection?

**Response:** This is a great question. The boundaries are represented as facets (i.e., planar). And yes, it is resolved to approximately voxel size, with limited smoothing related to the level-set function that differentiates the snow from the air along the voxel boundary.

Line 147-148: I assume the particle orientation is also random, but please clarify.

**Response:** We have removed all methods related to particles, so this section has been rewritten in it's entirety.

Line 169: Perhaps I missed it, but how many dTheta bins are used for the calculation of p(cos(Theta))?

**Response:** We used 180 bins, but this is a user configurable option.

Line 176: Grammatical issue.
**Response:** Corrected, thank you.

Line 202-203: The meaning of this sentence ("... instead of...") is unclear to me. What is the distinction between these two methods?

**Response:** In this method, the energy absorption is computed directly from the absorption coefficient of ice and a mean distance traveled within ice between scattering events, whereas most commonly in medium models, the medium is assigned an absorption coefficient based on the complex refractive index and single-scattering albedo.

Equation 20: How many bins are used to compute the DCRF?
**Response:** We used 10x10 degree bins following Kaempfer et al. 2007.

Line 257: "... constant reflectance of approximately 4%..." - Is the uncertainty of this reflectance known, and if so, how important is this uncertainty? A simple model sensitivity study (e.g., with +/- X% reflectivity) could shed light on how important this uncertainty is for interpretation of model-measurement comparison.

**Response:** We suspect that the uncertainty of this is rather small, we took numerous reflectance measurements of this panel throughout the data collection periods and found a majority of the uncertainty was concentrated within the water vapor and CO2 bands. In response to this comment, we have performed these runs with +- 5% uncertainty in the lower-boundary reflectance, and show negligible impacts, even at the shallowest snow.

Section 3.1: How were the mesh samples generated? Are these simply individual micro-CT scans, or were samples somehow stitched together to create the 800 mm^3 volumes? Also, it would be helpful to clarify (once again) at the beginning of this section that the 1-D model is used to simulate albedo, using optical properties generated from 3-D simulations of individual ice particles from the scans.

**Response:** These mesh samples were generated from individual micro-CT scans of approximately 2.0cm width and 10cm depth.  These are subsets of that to avoid assimilating edge pixels along the circumference of the micro-CT sample. We have reiterated how the 3D samples are used to generate the optical properties for 1D samples.

Line 337: By "optical thickness", I assume you mean the thickness of snow needed to achieve 5% transmittance, but it might help to apply more precise wording.

**Response:** Thank you for this suggestion, we have clarified this to read "transmittance depth" following a similar comment from another reviewer.

Figure 10 caption: Please note the snow thickness assumed in these model studies.

**Response:** Done, thank you.

References:

Carmagnola, C. M., Domine, F., Dumont, M., Wright, P., Strellis, B., Bergin, M., Dibb, J., Picard, G., Libois, Q., Arnaud, L., and Morin, S.: Snow spectral albedo at Summit, Greenland: measurements and numerical simulations based on physical and chemical properties of the snowpack, The Cryosphere, 7, 1139-1160, https://doi.org/10.5194/tc-7-1139-2013, 2013.

Dumont, M., Flin, F., Malinka, A., Brissaud, O., Hagenmuller, P., Lapalus, P., Lesaffre, B., Dufour, A., Calonne, N., Rolland du Roscoat, S., and Ando, E.: Experimental and model-based investigation of the links between snow bidirectional reflectance and snow microstructure, The Cryosphere, 15, 3921-3948, https://doi.org/10.5194/tc-15-3921-2021, 2021.

Grenfell, T. C., Warren, S. G., and Mullen, P. C.: Reflection of Solar Radiation by the Antarctic Snow Surface at Ultraviolet, Visible, and Near-Infrared Wavelengths, J. Geophys. Res., 99, 18,669-18,684, 1994.

Liou, K. N., Takano, Y., He, C., Yang, P., Leung, L. R., Gu, Y., and Lee, W. L.: Stochastic parameterization for light absorption by internally mixed BC/dust in snow grains for application to climate models, J. Geophys. Res.: Atmos., 119, 7616-7632, https://doi.org/10.1002/2014JD021665, 2014.

---

## Referee Report (RR1)

Second review of « A generalized photon-tracking approach to simulate spectral snow albedo and transmissivity using X-ray microtomography and geometric optics », by Theodore Lechter et al.

**General comments**

Much effort have been made to improve the scientific quality and the clarity of the manuscript, which has been significantly revised compared to the initial version. New figures have been appended, and many others were updated. This is appreciated and certainly makes this study more convincing. However I believe an additional effort is still required to make it an even more significant and scientifically robust contribution. In particular, issues that have been partly tackled in the response to reviewers should be further tackled and included in the paper. These issues are detailed below.

**Specific comments**

1) The most questioning remaining issue concerns the definition of extinction coefficient for a porous medium. It is explained in the text (and in more details in the response to reviewers) that the computed extinction coefficient does not perfectly match the expected dependence (from theory on particulate media) on SSA and density. This is attributed to the way extinction coefficient is computed, with photons launched from anywhere in the medium, and not only in the air (which is what is implicitly done when launching photons on an isolated ice particle in air). I believe this raises a very important question about the definition (if only it means anything) of the extinction coefficient for a porous medium. In particular with this method the authors find for a collection of spheres a value of the extinction coefficient different than the commonly accepted value. What is the meaning of having different extinction coefficients, when such a quantity should be intrinsic to the medium? This critical question cannot be overlooked, otherwise your method could be erroneously replicated by others. My guess is that defining extinction coefficient and phase function (and probably absorption along extinction paths along with those two quantities) for a porous medium is not univocal, and definitely not trivial. Clearly there exist alternative strategies that result in different values. However I think that for radiative transfer what matters is the product $\gamma_{ext}\cdot(1-g)$, which is probably the same for any strategy (also explaining why the computed albedos satisfactorily match). Hence I would recommend to estimate $\gamma_{ext}$ in the common way (launching photons from the air phase only) and to compute the phase function using the deviation between entrance angle (into the ice phase) and final escape angle (after some internal reflections). Applying this to a collection of spheres would help check whether accepted values are retrieved for the phase function (or g) and extinction coefficient, and whether both strategies are indeed equivalent for radiative transfer. Actually you can already check whether $\gamma_{ext}\cdot(1-g)$ obtained with your method for the collection of spheres matches the "particle approach" value. This is certainly beyond the initial scope of this paper, but the strategy proposed cannot really be accepted before it is proved that it replicates exactly (regarding this, care should be taken to ensure that the obtained match is convincing, which was not the case in the new figure of spectral albedos provided in the response to reviewers) what is expected for a collection of spheres. Tackling this issue would strongly support the presented strategy, and would be a major contribution to radiative transfer in snow (actually in any weakly absorbing porous medium).

2) Some figures (e.g. spectral albedo in Fig. 13 (now Fig. 10)) and values (e.g. Table 1) have changed between both versions, while they apparently correspond to the same measurements, which is puzzling. It would be helpful to explain these differences (at least for the reviewers).

**Technical comments** (lines correspond to the track changes version)

l.16: the RMSE value might not be meaningful here, because it depends on the number of samples considered, on the characteristics of snow, on, the spectral range of the measurements… which are not detailed. Also, reflectance is rather "measured with a spectroradiometer" than "estimated".

l.92: this long paragraph is surprising because it already contains much information, especially about the second component of the model. Shouldn't a large part of it (after l.98) be moved to the subsections?

l.95: should the term "ray-tracing" or something equivalent appear here? Actually both models are somehow based on ray-tracing

l.113: could you expand on why "the semi-quantized approach described here reduces the number of photons required to achieve a statistically robust result"

l.116: "in this model": is it in the first or second component that phase and diffraction are ignored? I'd rather say in the first component, since the second component just takes as inputs statistically representative single scattering properties, no matter where they come from

l.136: isn't "a photon of light" redundant?

l.140: ice-path fraction or mean path fraction traveled within ice? Maybe chose a single consistent term.

l.143: the choice of starting anywhere in the medium (ice or air) seems (according to the author comments) to be the reason for not matching the expected relationship between extinction coefficient, SSA and density. Starting the paths only in the air would definitely change the obtained extinction coefficient (and the phase function accordingly). So it raises a fundamental question about how to define the extinction coefficient of a porous medium, which may be an ill-posed question. See specific comment 1).

l.215: consider providing here (or in the caption of Figure 2)  the number of photons used to compute these statistics. Also the exponential fit does not seem very convincing. Discussing errors in this fitting procedure (fitting for instance the log of the POE) would be useful. To which extend could this uncertainty of the fit explain discrepancies with the usual dependency on density and SSA?

Figure 1: a) ke in the legend should have a unit. By the way what is ke? In the Figure caption refer to a, b, c, d. c) what is the dashed red line? d) 1000 nm should be in the suptitle rather than in the legend

l.260: still, I think this Eq. (17) implies that the total physical length traveled by the photons is the sum of the s segments, while due to the internal reflections quantified by the B parameter a longer total distance is traveled, so that s should be scaled accordingly

l.280: Figure 2

l.339: one tenth rather than 10 times?

Figure 5 caption: different types of what?

Tables 1 and 2: could the g values (of the computed phase function) be indicated as well?

l.405: is it obvious what an exponential increase means? Maybe provide the functional form of the dependence

l.412: the definition of transmittance is ambiguous here. "Within a snowpack" suggests that the snowpack is thick and that the downward flux is estimated at an intermediate depth, rather than at the bottom of the snowpack. Please clarify this, because both quantities (e.g. flux below a 5 cm layer and flux at 5 cm depth in a thick layer) are very different.

Table 2: depth should start at 0 at the surface no?

l.474: it would be worth commenting the fact that the scaling of Eq. (21) is not 0.5 as would be expected from the studies cited above

l.477: not clear why rounded grains are supposed to have highest B values. Spherical particles have a low B compared to fresh, supposedly fractal snow

l481: as both $\gamma_{ext}$ and $F_{ice}$ depend on density, what is the meaning of varying them independently?

Figure 13: extinction coefficient should have units mm$^{-1}$. What do the points correspond to?

l.496: was it clearly stated how $F_{ice}$ depends on B (when density is fixed)?

l.531: it should be made clear that the refractive index is taken constant only for the optical properties computations, not for the absorption modeling

l.532: another point regarding the MIR is that radiative transfer should not be applied this way to dense media (shadowing effects are ignored while they are obvious when snow grains touch each other). It's actually a chance that the dilute media theory applies that well to snow (see Kokhanovsky (2004) for more details) but this holds only as long as snow is weakly absorbing, which is not the case anymore beyond 1400 nm. Likewise, surface features (the topmost few mm) matter in this spectral range, while they are probably not captured by manual measurements of snow.

l.546: the term Monte Carlo is ambiguous because to me the estimation of the optical properties is also based on a Monte Carlo method (ray-tracing with various possible paths)

l.556: it's not clear why this work would help with subnivean hazards. In the introduction or here it would be nice to clarify what kind of features could be seen (or not) through the snow, and why knowing the snow transmittance can help

**Reference**:

Kokhanovsky, A. A. (2004). *Light scattering media optics*. Springer Science & Business Media.

---

## Referee Report (RR2)

Third review of « A generalized photon-tracking approach to simulate spectral snow albedo and transmissivity using X-ray microtomography and geometric optics », by Theodore Lechter et al.

**General comments**

I thank the authors for the significant effort made to revise the manuscript, in particular for the critical discussion on the meaning of scattering coefficient for a porous medium, and for changing their method to compute this quantity from snow samples. I believe the updated discussion is very valuable for the overall quality of the paper and raises issues that would need to be considered in the future, even though I fully agree this is beyond the scope of the present study. The paper can now be published in *The Cryosphere*. Below some final technical issues are pointed out, that can be easily fixed. The most critical is about accounting or not for diffraction when computing the scattering coefficient and $g$ (the choice should be consistent).

**Technical comments** (lines correspond to the track changes version)

l.87: do not change paragraph

l.91: idem, include all this in the same first paragraph of the section

l.104: "multiple scattering within the two-phase medium" is unclear here. Consider removing that, because independent scattering mostly refers to "no interference"

l.107 the subscript "ext" should not be italic. True everywhere, and for "sca", "ice" etc.

l.107-109: the variables for the physical quantities should not be in parenthesis

l.139: weird to see again Snell's and Fresnel's laws (introduced a few lines earlier). Maybe move to a few lines earlier. If the manuscript is too long, these few formulas (known to most of the readers) could also be put in an Appendix.

l.163: does $d_i$ includes the lengthening of paths due to internal reflections (effect of B)? Or is it the length of the straight line between scattering events? The text suggests option 1, is it what you meant and what is done by Randrianalisoa and Baillis (2010)? I think the usual definition of extinction coefficient would rather use the straight line between scattering events (e.g. Eq. 7 of Malinka, 2014)

l.180: not clear how B and $F_{ice}$ (why not just B and density?) are then used. Clarify here or later (in the RT code) when more relevant

l.212: the reference should be in parenthesis

l.235: diffuse radiation means isotropic which is not random, it corresponds to a well-defined angular distribution

Figure 2: panel a is more a cumulated distribution than a distribution, no? The suptitle is unclear, consider removing it. Also, maybe put the sample first (a) and the optical properties then (switch a and b). In the definition of g isn't the phase function missing? Important point: it seems that diffraction is included in the computation of g, hence considered as scattering. But it is apparently not included in the estimation of the scattering coefficient which relies on geometrical paths. Can you clarify this, and update the estimation of g (using only the geometrical part) if needed. Although it may not impact your results since you don't directly use g, but the full phase function instead.

l.261 : in snow with

l.430: should be "ext", and SSA should not be italic

l.480: larger than

l.490: have a

---

## Author Response (AR2)

Second review of « A generalized photon-tracking approach to simulate spectral snow albedo and transmissivity using X-ray microtomography and geometric optics », by Theodore Lechter et al.

**General comments**

Much effort have been made to improve the scientific quality and the clarity of the manuscript, which has been significantly revised compared to the initial version. New figures have been appended, and many others were updated. This is appreciated and certainly makes this study more convincing. However I believe an additional effort is still required to make it an even more significant and scientifically robust contribution. In particular, issues that have been partly tackled in the response to reviewers should be further tackled and included in the paper. These issues are detailed below.

**Specific comments**

1) The most questioning remaining issue concerns the definition of extinction coefficient for a porous medium. It is explained in the text (and in more details in the response to reviewers) that the computed extinction coefficient does not perfectly match the expected dependence (from theory on particulate media) on SSA and density. This is attributed to the way extinction coefficient is computed, with photons launched from anywhere in the medium, and not only in the air (which is what is implicitly done when launching photons on an isolated ice particle in air). I believe this raises a very important question about the definition (if only it means anything) of the extinction coefficient for a porous medium. In particular with this method the authors find for a collection of spheres a value of the extinction coefficient different than the commonly accepted value. What is the meaning of having different extinction coefficients, when such a quantity should be intrinsic to the medium? This critical question cannot be overlooked, otherwise your method could be erroneously replicated by others. My guess is that defining extinction coefficient and phase function (and probably absorption along extinction paths along with those two quantities) for a porous medium is not univocal, and definitely not trivial. Clearly there exist alternative strategies that result in different values. However I think that for radiative transfer what matters is the product $\gamma_{ext} \cdot (1-g)$, which is probably the same for any strategy (also explaining why the computed albedos satisfactorily match). Hence I would recommend to estimate $\gamma_{ext}$ in the common way (launching photons from the air phase only) and to compute the phase function using the deviation between entrance angle (into the ice phase) and final escape angle (after some internal reflections). Applying this to a collection of spheres would help check whether accepted values are retrieved for the phase function (or g) and extinction coefficient, and whether both strategies are indeed equivalent for radiative transfer. Actually you can already check whether $\gamma_{ext} \cdot (1-g)$ obtained with your method for the collection of spheres matches the "particle approach" value. This is certainly beyond the initial scope of this paper, but the strategy proposed cannot really be accepted before it is proved that it replicates exactly (regarding this, care should be taken to ensure that the obtained match is convincing, which was not the case in the new figure of spectral albedos provided in the response to reviewers) what is expected for a collection of spheres. Tackling this issue would strongly support the presented strategy, and would be a major contribution to radiative transfer in snow (actually in any weakly absorbing porous medium).

We thank the reviewer for this thoughtful and engaging question. In response to this question, we undertook a major effort to track down the discrepancies between various approaches and ultimately reassessed how to get the extinction coefficient for our model. The outcome of this is summarized here.

1. First, a limited assessment comparing the inverse-transport length [$\gamma_{ext}*(1-g)$] computed for a selection of spheres with photons initialized in the air phase only and the asymmetry parameter from the "whole particle" scattering does indeed match with $\gamma_{ext}$ computed with photons initialized in both phases and the "localized" scattering. This result would support the hypothesis posed in the discussion part of the previous draft (see figure below).

[Figure]

2. A more robust comparison of a variety of different methods for computing $\gamma_{ext}$ through ray-tracing methods and an additional literature review suggest that:

a. The Xiong et al. (2015) method of initializing particles at random positions throughout the medium is not entirely consistent with the definition of the extinction (scattering) coefficient as the inverse of the mean distance traveled between scattering events. In essence, because the particles are not by necessity initialized at a scattering event (particle boundary), the extinction coefficient is artificially high following this method. This is explicitly stated in Randrianaliosa and Ballis (2010). Further, through additional work, we were able to determine that the curve fitting used by Xiong et al. (2015) is largely unnecessary, as it will converge to generate a $\gamma_{ext}$ that is equal to simply 1/mean free path.

b. In following the points illustrated in (a), we have decided that the Xiong et al. (2015) method for determining the extinction coefficient has some potential issues with it that we hadn't fully grasped until diving deeply into it, and while it may work in some instances, we think it would be better to choose a more reliable method. Accordingly, we have decided to compute the extinction coefficient following the method in Randrianaliosa and Ballis (2010), which simply computes the extinction (scattering) coefficient from the mean path length between scattering events during the ray-tracing step. In this method, scattering events are defined with respect to the whole particle, such that internal reflections are not counted as discrete events. Scattering events only occur when the ray enters or exits the particle via either reflection or transmission. We feel that this method is more robust and more consistent with the medium based model. Further, in a comparison of different methods for computing $\gamma_{ext}$, we find that the Randrianaliosa and Ballis approach reproduces the SSA*rho/4 better than other methods. (see below scatter plot)

[Figure]

*Scatter plot of $\gamma_{ext}$ for various ray-tracing techniques plotted against theory.*
*Note that the Xiong et al. technique here is for particles initialized in the air-phase.*

    c.   Because this framework defines scattering events with respect to the whole particle, we have accordingly opted to use to the "whole particle" phase function as well. This is computed using the entire sample, such that segmenting grains is not required.

    d.   All of the methods compared in this latest round of reviews (Xiong et al. 2015; Malinka et al., 2014; Randrianaliosa and Ballis, 2010) produce a large spread in results (see above), showcasing the difficulty in estimating optical properties through ray-tracing techniques. While tracking down the reasons for this would be a worthwhile endeavor, we feel that doing so is beyond the scope of this work. We speculate that there are several nuanced differences between the methods that are responsible for the differences, e.g.: random vs. controlled transport paths, the role of total-internal-reflection traps. However, we haven't performed a more in-depth investigation into this as of yet. We do, however, note that others have remarked on some of the uncertainty in understanding and estimating the extinction coefficient for snow (e.g., Malinka et al. 2014).

    e.   All in all, we think that a large source of this uncertainty that we have struggled with is because the Xiong et al. (2015) method treats scattering events differently than other approaches, with respect to how both the mean free path and extinction coefficient and phase function are treated. We would like to note that when updating our model such that scattering events are more similar to the most common understanding, the changes to our results in the plane-parallel were almost negligible. So we speculate that the Xiong et al. 2015 method may be fundamentally consistent with other approaches. However, we are unable make this claim definitively, and have chosen not to as part of this work due to significant and non-intuitive/unexpected nuances involved on the

computer-science side. Accordingly, we have opted to stick with more accepted definitions. We have added a paragraph to the discussion on this point and indicated that this model can be used to further investigate.

2) Some figures (e.g. spectral albedo in Fig. 13 (now Fig. 10)) and values (e.g. Table 1) have changed between both versions, while they apparently correspond to the same measurements, which is puzzling. It would be helpful to explain these differences (at least for the reviewers).

Response: Not providing an explanation for this change in the first round of revisions was an oversight on my part during the first round of reviews. In short, we had identified a minor coding error in our script used to plot the spectral albedo that pointed to incorrect reference scans for the 4.5 and 2.5 cm albedos. We also made a minor downward adjustment to the reflectance value of the reference panel as a majority of clean-snow reflectance's exceeded 1 in the blue-visible spectrum. To this point, all code and observational measurement and metadata used to make these figures will be included with the code repository on GitHub.

**Technical comments** (lines correspond to the track changes version)

l.16: the RMSE value might not be meaningful here, because it depends on the number of samples considered, on the characteristics of snow, on, the spectral range of the measurements… which are not detailed. Also, reflectance is rather "measured with a spectroradiometer" than "estimated".

**Response:** Thank you, we have revised the text to include the spectral range of the measurements for greater context. The remaining information is detailed in the body of the article. The recommended change in wording from "estimated" to "measured" was also made.

l.92: this long paragraph is surprising because it already contains much information, especially about the second component of the model. Shouldn't a large part of it (after l.98) be moved to the subsections?

**Response:** Thank you for this suggestion.  We have shortened this paragraph and merged most of the information contained within it into the sub-section describing, in detail, the plane parallel model.

l.95: should the term "ray-tracing" or something equivalent appear here? Actually both models are somehow based on ray-tracing

Response: Thank you for the suggestion, this has been reworded as part of the l.92 response.

l.113: could you expand on why "the semi-quantized approach described here reduces the number of photons required to achieve a statistically robust result"

**Response:** We had used the phrase "semi-quantized" to help specify that photon-tracks were considered discrete (i.e., quantized) while the absorption was treated continuously.  However, in some of the rewording/reorganizing of the paper, we have removed this phrasing, so it is no longer an issue.

l.116: "in this model": is it in the first or second component that phase and diffraction are ignored? I'd rather say in the first component, since the second component just takes as inputs statistically representative single scattering properties, no matter where they come from

**Response:** Thank you for this clarification. To be more specific, we have rephrased this sentence as: "*One critical simplification we make in determining the optical properties of a given snow sample is that we ignore the wave properties of light …*"

l.136: isn't "a photon of light" redundant?

**Response:** Yes, good catch. We have revised.

l.140: ice-path fraction or mean path fraction traveled within ice? Maybe chose a single consistent term.

**Response:** Thank you for the suggestion, we have replaced "mean path fraction within ice" with ice-path fraction.

l.143: the choice of starting anywhere in the medium (ice or air) seems (according to the author comments) to be the reason for not matching the expected relationship between extinction coefficient, SSA and density. Starting the paths only in the air would definitely change the obtained extinction coefficient (and the phase function accordingly). So it raises a fundamental question about how to define the extinction coefficient of a porous medium, which may be an ill-posed question. See specific comment 1).

**Response:** This is a question that we have been grappling with since going over the initial reviews.

We refer you to our more in-depth response to comment 1. However, we do reiterate here that there is significant ambiguity regarding the definition of the scattering/extinction coefficient in porous media. In particular, scattering events can be defined either as occurring over a whole particle, or at the dielectric boundary comprising the particle surface. While traditional RT methods and theory accept the former, ray-tracing is well suited to the latter, and we were not able to find any published research that truly attempted to reconcile these approaches. The method from Xiong et al. 2015, follows the latter method, which is one reason we speculate that it does not reproduce theory under idealized conditions. Furthermore, this method, whereby particles are initialized *at random* throughout the medium as opposed to on air/ice boundaries is explicitly discounted as wrong by Randrianalisoa and Baillis (2010). Further, Malinka et al. (2014) suggests that the definition of the extinction coefficient is not entirely settled and that there is no rigorous proof that the forms in Van De Hulst (1957) and Kokhanovsky and Zege (2004) hold true for random mixtures and dense-packed media. So, while there definitively seem to be fundamental unanswered questions regarding scattering in the extinction coefficient in porous media, we feel that trying to answer them here is beyond the scope of this work. Accordingly, have reverted to a method that is most consistent with accepted theory due to outstanding issues and growing skepticism towards the Xiong et al. (2015) method that we had used initially. We do want to note that while the optical properties generated from the Xiong et al. (2015) method are quite different from the optical properties generated from other methods, the actual simulated spectral albedos computed from each set are almost indistinguishable, at least for the simulations discussed in this study.

l.215: consider providing here (or in the caption of Figure 2) the number of photons used to compute these statistics. Also the exponential fit does not seem very convincing. Discussing errors in this fitting procedure (fitting for instance the log of the POE) would be useful. To which extend could this uncertainty of the fit explain discrepancies with the usual dependency on density and SSA?

**Response:** We have included the number of photons in the caption for figure 2. We have included a response made to a similar comment from another reviewer on the first round of revisions that discusses some of the uncertainty in the curve fitting procedure below. While the actual uncertainty in the curve fitting procedure is generally small, it doesn't always match the POE. However, in light of the response to the first major comment, the curve fitting procedure is moot, since it is no longer used.

**Previous Reviewer Response:** The uncertainty for a given value of the extinction coefficient is very low for set parameters. For example, repeating the calculation of it for the same sample, same wavelength, and same curve fitting technique yields a standard deviation of <0.05 if enough photons are used (>2000 seems to be sufficient for convergence). However, there is more substantive uncertainty if the parameters that influence the curve fitting are modified. For example, if the distance sampling is different, or if the initial guess that feeds into the curve fit is modified can yield differences as high as 0.3.

Figure 1: a) ke in the legend should have a unit. By the way what is ke? In the Figure caption refer to a, b, c, d. c) what is the dashed red line? d) 1000 nm should be in the suptitle rather than in the legend

**Response:** Thank you for catching some of these inconsistencies from previous revisions. We have made all of the suggested updates to the figure and the figure caption. We removed the 1000nm remark entirely, and instead note in the methods section that we assume an index of refraction of 1.30.

l.260: still, I think this Eq. (17) implies that the total physical length traveled by the photons is the sum of the s segments, while due to the internal reflections quantified by the B parameter a longer total distance is traveled, so that s should be scaled accordingly

**Response:** We think that this question is part of the same ambiguity surrounding how scattering events are defined and quantified with the extinction coefficient and scattering phase function highlighted in previous comments. For instance, if internal scattering events are considered distinct when computing the extinction coefficient, an internal path extension would not be required. However, now that our methods are more closely aligned with currently accepted definitions, it is

clearer that an adjustment to Fice is needed to account for this. In revisiting this from an analytical standpoint, we can show by combining equations from Libois et al. 2014 and Libois et al. 2019 that the absorption coefficient can be expressed in terms of Fice as -> $\sigma = F_{ice}\left(1 + \frac{\rho_s}{\rho_{ice}}[B - 1]\right)\gamma$. Accordingly, the scaling of the "s" length for equation 17 for absorption when computing it through $F_{ice}$ is close to 1 (1.02 - 1.15) for most samples. We have added this scaling to the absorption function used in the plane parallel model, and more explicitly described how the "B" parameter is computed, since it is now a critical optical property used in the plane-parallel model. However, we do note that there are some outstanding uncertainties surrounding this parameter that seem to be related to the fact that it's being calculated within a sample comprised of a collection of snow grains, rather than a single particle. We have included some text on this in the discussion.

l.280: Figure 2

Response: Fixed, thank you.

l.339: one tenth rather than 10 times?

Response: Thank you for catching this typo, we have revised the text to one tenth.

Figure 5 caption: different types of what?

Response: Thank you for catching this typo, we have completed the sentence with "...types of snow grains."

Tables 1 and 2: could the g values (of the computed phase function) be indicated as well?

Response: Yes, see updated tables.

l.40: is it obvious what an exponential increase means? Maybe provide the functional form of the dependence

**Response:** We have replaced this sentence with: ***"This analysis shows an increase in albedo at high zenith angles that is most pronounced in the NIR that represents the functional dependence between albedo and cos(θ). This result is broadly consistent with results from previous studies that compare snow albedo and zenith angle."***

l.412: the definition of transmittance is ambiguous here. "Within a snowpack" suggests that the snowpack is thick and that the downward flux is estimated at an intermediate depth, rather than at the bottom of the snowpack. Please clarify this, because both quantities (e.g. flux below a 5 cm layer and flux at 5 cm depth in a thick layer) are very different.

Response: We have clarified this by rephrasing the sentence on Line 323 to say: ***"To accomplish this, the optical properties of the µCT samples in Fig. 7 are used to simulate and compare the downward flux at varying depths (Fig. 11)"***

Table 2: depth should start at 0 at the surface no?

**Response:** This is a bit tricky, in that while we agree that "depth" with the snow surface set as 0 is technically correct, we often think of it as increasing from the ground as zero when we refer to snow depth. Accordingly, we would like to leave this as is.

l.474: it would be worth commenting the fact that the scaling of Eq. (21) is not 0.5 as would be expected from the studies cited above

**Response:** There is a detailed discussion of this in the Discussion section of the manuscript. See previous comments regarding the extinction coefficient and updated results. While we are not getting exactly ¼*SSA*rho, the new mean-free-path method is much closer.

l.477 : not clear why rounded grains are supposed to have highest B values. Spherical particles have a low B compared to fresh, supposedly fractal snow.

**Response:** Thank you for this comment. In this instance, while we were able to sample fresh snow, we did not measure a good sampling of dendritic or fractal grains with the microCT. However, we note that the relationship between grain form and B was not clear, and that the clearest relationship was to see higher B values for smaller grains, which happened to be round (or at least roundish) as a majority of the larger grains were faceted or broken. In the ray tracing framework employed here, we find that photons are consistently more likely to get "trapped" in total internal reflection with smaller rounded particles, than they are for larger, more irregularly shaped particles, which we suspect increases the internal path length. We have rephrased this sentence to be clearer with respect to what our results show within the context of previous work:

***"We note that there is no significant relationship between B and snow grain form or size, however there is a general tendency for B to be highest for samples with higher SSA and smaller grains which is qualitatively consistent with Kokhanovsky and Zege (2004) and Libois et al. (2014). However, due to limitations in the variety of snow samples and MicroCT resolution we are unable to make any concrete conclusions regarding relationships between B and grain form."***

l.481 (367): as both $\gamma_{ext}$ and $F_{ice}$ depend on density, what is the meaning of varying them independently?

**Response:** The purpose of this was to assess the relative sensitivity of transmittance to each property independently. While certainly they are somewhat linked through snow density, we think it is important to include this sensitivity to illustrate that a) the transmittance depth is more sensitive to the extinction/scattering coefficient, and b) that the sensitivity to Fice is also dependent on the scattering coefficient (e.g., it is more sensitive to Fice for a low extinction coefficient). Also, because the extinction coefficient is also a function of SSA, it seems plausible that these combinations of $\gamma_{ext}$ and $F_{ice}$ could exist.

Figure 13: extinction coefficient should have units $mm^{-1}$. What do the points correspond to?

Response: Thank you for this catch, we have corrected the figure caption. Each point corresponds to a sample within that snow layer. We have added additional text in the figure caption to clarify this.

l.496: was it clearly stated how $F_{ice}$ depends on B (when density is fixed)?

**Response:** We are not sure what statement this refers to since the line numbers provided do not align with those in the submitted manuscript. Without additional context, it is difficult to discern what the reviewer is referring to. However, we have included some additional discussion regarding the relationship between B and Fice that shows this dependence.

l.531: it should be made clear that the refractive index is taken constant only for the optical properties computations, not for the absorption modeling

Response: Done, thank you.

l.532: another point regarding the MIR is that radiative transfer should not be applied this way to dense media (shadowing effects are ignored while they are obvious when snow grains touch each other). It's actually a chance that the dilute media theory applies that well to snow (see Kokhanovsky (2004) for more details) but this holds only as long as snow is weakly absorbing, which is not the case anymore beyond 1400 nm. Likewise, surface features (the topmost few mm) matter in this spectral range, while they are probably not captured by manual measurements of snow.

Response: Thank you for bringing up these additional points. We have added text to the discussion covering these additional sources of uncertainty.

l.546: the term Monte Carlo is ambiguous because to me the estimation of the optical properties is also based on a Monte Carlo method (ray-tracing with various possible paths)

Response: We have revised the text to the following for clarification: ***"A primary goal of this modeling approach is to expand upon previous approaches aimed at incorporating 3D renderings of real snow microstructure into radiative transfer models for snowpacks of arbitrary depth, while maintaining the ray tracing methods utilized in the original Kaempfer et al. (2007) model."***

l.556: it's not clear why this work would help with subnivean hazards. In the introduction or here it would be nice to clarify what kind of features could be seen (or not) through the snow, and why knowing the snow transmittance can help

Response: We have modified this phrasing to remove subniviean hazards in replace it with: ***"improved capabilities for determining the optical properties of shallow snowpack".***

**Reference**:

Kokhanovsky, A. A. (2004). *Light scattering media optics*. Springer Science & Business Media.

Randrianalisoa, J. and Baillis, D.: Radiative properties of densely packed spheres in semitransparent media: A new geometric optics approach, Journal of Quantitative Spectroscopy and Radiative Transfer, 111, 1372–1388, 2010.

Malinka, A. V.: Light scattering in porous materials: Geometrical optics and stereological approach, Journal of Quantitative Spectroscopy and Radiative Transfer, 141, 14–23, 2014.

---

## Author Response (AR3)

Third review of « A generalized photon-tracking approach to simulate spectral snow albedo and transmissivity using X-ray microtomography and geometric optics », by Theodore Lechter et al.

**General comments**

I thank the authors for the significant effort made to revise the manuscript, in particular for the critical discussion on the meaning of scattering coefficient for a porous medium, and for changing their method to compute this quantity from snow samples. I believe the updated discussion is very valuable for the overall quality of the paper and raises issues that would need to be considered in the future, even though I fully agree this is beyond the scope of the present study. The paper can now be published in *The Cryosphere*. Below some final technical issues are pointed out, that can be easily fixed. The most critical is about accounting or not for diffraction when computing the scattering coefficient and *g* (the choice should be consistent).

**Technical comments** (lines correspond to the track changes version)

l.87: do not change paragraph
Response: We agree and have updated text accordingly.

l.91: idem, include all this in the same first paragraph of the section
Response: Same as previous response.

l.104: "multiple scattering within the two-phase medium" is unclear here. Consider removing that, because independent scattering mostly refers to "no interference"
Response: Done, thank you for the suggestion.

l.107 the subscript "ext" should not be italic. True everywhere, and for "sca", "ice" etc.
Response: Thank you, we have fixed the subscripts throughout.

l.107-109: the variables for the physical quantities should not be in parenthesis
Response: These have also been fixed throughout.

l.139: weird to see again Snell's and Fresnel's laws (introduced a few lines earlier). Maybe move to a few lines earlier. If the manuscript is too long, these few formulas (known to most of the readers) could also be put in an Appendix.
Response: We have reworded such that the first paragraph is kept as a more general overview and the second paragraph explicitly states the laws and introduces the equations.

l.163: does $d_i$ includes the lengthening of paths due to internal reflections (effect of B)? Or is it the length of the straight line between scattering events? The text suggests option 1, is it what you meant and what is done by Randrianalisoa and Baillis (2010)? I think the usual definition of extinction coefficient would rather use the straight line between scattering events (e.g. Eq. 7 of Malinka, 2014).

Response: Considering that there are several different methods for determining the scattering coefficient from microCT and ray-tracing methods, it's not clear that there is a "usual" or universally accepted method for determining the extinction coefficient. We choose to use the Randrianalisoa and Baillis (2010) method here because it a) fits best within the ray-tracing methods used in the study, and b) produced values most consistent with the theoretical rho*SSA/4 out of all the different methods we tested. Accordingly, we would like to retain the current method of computing the extinction coefficient and continue to pursue differences in various methods as part of ongoing research. To that end, this method would include the impact of path lengthening to a degree, as the scattering coefficient is equal to the TOTAL pathlength / number of scattering events, where each "particle" results in a single scattering event, whether it be reflection at the particle surface, or transmission through the far-side of the particle.

l.180: not clear how B and $F_{ice}$ (why not just B and density?) are then used. Clarify here or later (in the RT code) when more relevant

Response: We think it is appropriate to include the path-length extension with the absorption in this framework due largely to the finding that the Randrianalisoa and Baillis (2010) method fits expected theory well. We also can show that for the examples in this paper, there is minimal impact to the results if a path-length extension is not used. For clarity, we have removed this specific reference to how B and Fice are used on l.180, and instead focus on a detailed description of how absorption is computed in the model in the section 2.2 where medium model is described.

l.212: the reference should be in parenthesis
Response: Corrected, thank you.

l.235: diffuse radiation means isotropic which is not random, it corresponds to a well-defined angular distribution
Response: Thank you for this clarification, we have reworded this to try and be more specific ->

*"This initial direction can be prescribed as downward pointing with a uniformly random zenith and azimuth angles representing isotropic diffuse radiation, or ..."*

Figure 2: panel a is more a cumulated distribution than a distribution, no? The suptitle is unclear, consider removing it. Also, maybe put the sample first (a) and the optical properties then (switch a and b). In the definition of g isn't the phase function missing? Important point: it seems that diffraction is included in the computation of g, hence considered as scattering. But it is apparently not included in the estimation of the scattering coefficient which relies on geometrical paths. Can you clarify this, and update the estimation of g (using only the geometrical part) if needed. Although it may not impact your results since you don't directly use g, but the full phase function instead.

Response: We have modified the figure according to these suggestions. THANK YOU for noticing that our definition for g was missing the phase function, we completely missed that mistake during proof reading. We chose to include diffraction in representing g for the following reasons: a) We can't recall ever seeing a geometric-only asymmetry parameter reported in the literature and we think that including diffraction in the computation places the values in better context with current literature, b) By including a "full" asymmetry parameter, the model optical properties could be more easily used in other two-stream RT models (e.g., TARTES), and c) As you alluded to in your comment, we do not use g in any of the other model components, so including diffraction as part of its computation does not improperly mix assumptions in the model. For clarity, we have modified our tables to include both the geometric and full "g" parameters.

l.261 : in snow with

Response: Done

l.430: should be "ext", and SSA should not be italic

Response: Done

l.480: larger than

Response: Done

l.490: have a

Response: Done